# The ACF chromatin-remodeling complex is essential for Polycomb repression

Elizabeth T Wiles*[†], Colleen C Mumford, Kevin J McNaught[‡], Hideki Tanizawa, Eric U Selker*

Institute of Molecular Biology, University of Oregon, Eugene, United States

**Abstract** Establishing and maintaining appropriate gene repression is critical for the health and development of multicellular organisms. Histone H3 lysine 27 (H3K27) methylation is a chromatin modification associated with repressed facultative heterochromatin, but the mechanism of this repression remains unclear. We used a forward genetic approach to identify genes involved in transcriptional silencing of H3K27-methylated chromatin in the filamentous fungus *Neurospora crassa*. We found that the *N. crassa* homologs of ISWI (NCU03875) and ACF1 (NCU00164) are required for repression of a subset of H3K27-methylated genes and that they form an ACF chromatin-remodeling complex. This ACF complex interacts with chromatin throughout the genome, yet association with facultative heterochromatin is specifically promoted by the H3K27 methyltransferase, SET-7. H3K27-methylated genes that are upregulated when *iswi* or *acf1* are deleted show a downstream shift of the +1 nucleosome, suggesting that proper nucleosome positioning is critical for repression of facultative heterochromatin. Our findings support a direct role of the ACF complex in Polycomb repression.

**\*For correspondence:**
tish.wiles@gmail.com (ETW);
selker@uoregon.edu (EUS)

**Present address:** [†]Department of Molecular Biology and Biochemistry, University of California, Irvine, United States; [‡]Genapsys, Inc, Westminster, United States

## Editor's evaluation

This manuscript provides strong evidence that ACF directly functions to promote Polycomb-dependent repression through chromatin remodeling, which has not been demonstrated. In addition, PRC2/H3K27me-dependent ACF targeting is novel. Finally, the authors' model that facultative chromatin can be classified into several groups based on their dependence on SET7, ASH1, and ACF (Figure 6) is potentially important for guiding future research directions of the field.

## Introduction

Polycomb repressive complex 2 (PRC2) methylates lysine 27 of histone H3 (H3K27), marking facultative heterochromatin (*Müller et al., 2002*; *Margueron and Reinberg, 2011*). Facultative heterochromatin contains regions of the genome that must remain transcriptionally plastic in order to respond to developmental or environmental cues (*Wiles and Selker, 2017*). Although H3K27 methylation has been established as a repressive chromatin modification, the mechanisms of repression are not fully understood (*Margueron and Reinberg, 2011*; *Ridenour et al., 2020*). One model for repression involves PRC1 binding to the H3K27 methyl-mark to facilitate chromatin compaction by self-association (*Grau et al., 2011*; *Cheutin and Cavalli, 2018*; *Boyle et al., 2020*). However, PRC1 is not present in all eukaryotes that bear H3K27 methylation-associated silencing, such as the filamentous fungus *Neurospora crassa* (*Jamieson et al., 2013*; *Schuettengruber et al., 2017*; *Wiles et al., 2020*), suggesting the existence of additional mechanisms of H3K27 methylation-associated repression.

In addition to histone modifications, nucleosome positioning may be critical for facultative heterochromatin function. The nucleosome, which is the fundamental unit of chromatin, consists of approximately 147 base pairs of DNA wrapped around an octamer of histone proteins (*Kornberg, 1974*;

**eLife digest** All the cells in an organism contain the exact same DNA, yet each type of cell performs a different role. They achieve this by turning specific genes on or off. To do this, cells wind their genetic code into structures called nucleosomes, which work a bit like spools of thread. Chemical modifications on these nucleosomes can determine whether a cell will use the genes spooled around it or not.

In many organisms, cells can turn genes off using a modification called H3K27 methylation. This mark attracts a protein complex called PRC1 that packs the genes away, making them inaccessible to the proteins that would activate them. But the filamentous fungus *Neurospora crassa* does not produce PRC1. This suggests that this organism must keep genes with the H3K27 mark switched off in a different way. One possibility is that H3K27 methylation somehow leads to changes in the position of nucleosomes on the genome, since having nucleosomes near the beginning of gene sequences can stop the cell from reading the code.

One protein complex responsible for positioning nucleosomes is known as the ATP-utilizing chromatin assembly and remodeling factor (ACF) complex, but it remained unknown whether it interacted with H3K27 methylation marks. To investigate further, Wiles et al. generated strains of *Neurospora crassa* that did not synthesize ACF and discovered that many of their genes, including ones marked with H3K27, were turned on. This was probably because the nucleosomes had shifted out of position, allowing the proteins responsible for activating the genes to gain access to the start of the genes' sequences.

Turning genes on and off at the right time is crucial for development, cell survival, and is key in tissues and organs working properly. Understanding the role of ACF adds to what we know about this complex process, which is involved in many diseases, including cancer.

*Luger et al., 1997*). Nucleosomes can be precisely positioned on DNA by ATP-dependent chromatin-remodeling proteins to produce a chromatin landscape that modulates accessibility for DNA transactions, such as transcription (*Lai and Pugh, 2017*). In particular, the precise positioning of the +1 nucleosome, the first nucleosome downstream of the transcription start site (TSS), is thought to be an important determinant of gene expression (*Rhee and Pugh, 2012*; *Nocetti and Whitehouse, 2016*). This dynamic nucleosome can occlude binding elements for transcriptional regulatory sites such as the TATA box (*Kubik et al., 2018*) and can serve as a barrier to RNA polymerase II (*Weber et al., 2014*).

In *Drosophila melanogaster*, the ATP-utilizing chromatin assembly and remodeling factor (ACF) complex (*Ito et al., 1997*) has been indirectly linked to the repression of Polycomb targets (*Fyodorov et al., 2004*; *Scacchetti et al., 2018*). The ACF complex is composed of the ATPase, Imitation Switch (ISWI), and the accessory subunit ACF1 (*Ito et al., 1999*). ACF is thought to act as a global nucleosome spacer and to contribute to repression genome-wide (*Baldi et al., 2018*; *Scacchetti et al., 2018*). Mutations in *Acf1* act as enhancers of *Polycomb* mutations and disrupt nucleosome spacing in facultative heterochromatin (*Fyodorov et al., 2004*; *Scacchetti et al., 2018*).

In order to improve our understanding of the control and function of facultative heterochromatin, we used forward genetics to identify factors required for silencing H3K27-methylated genes in *N. crassa*. As described here, this identified *iswi* (also known as *crf4-1*) and *acf1* (also known as *crf4-2*; *Itc1* in *S. cerevisiae*). We show that these proteins interact to form an ACF complex in *N. crassa*, that ACF is necessary for repression of a subset of H3K27-methylated genes, and that derepression is not simply due to loss of H3K27 methylation. ACF interacts with chromatin targets throughout the genome, yet specific interactions with H3K27-methylated regions are partly dependent on SET-7, the H3K27 methyltransferase. Finally, we show that when members of ACF are absent, H3K27-methylated genes that become upregulated display a specific downstream shift of the +1 nucleosome. Our findings support a model in which ACF remodels the chromatin landscape at H3K27-methylated regions of the genome to contribute to Polycomb silencing.

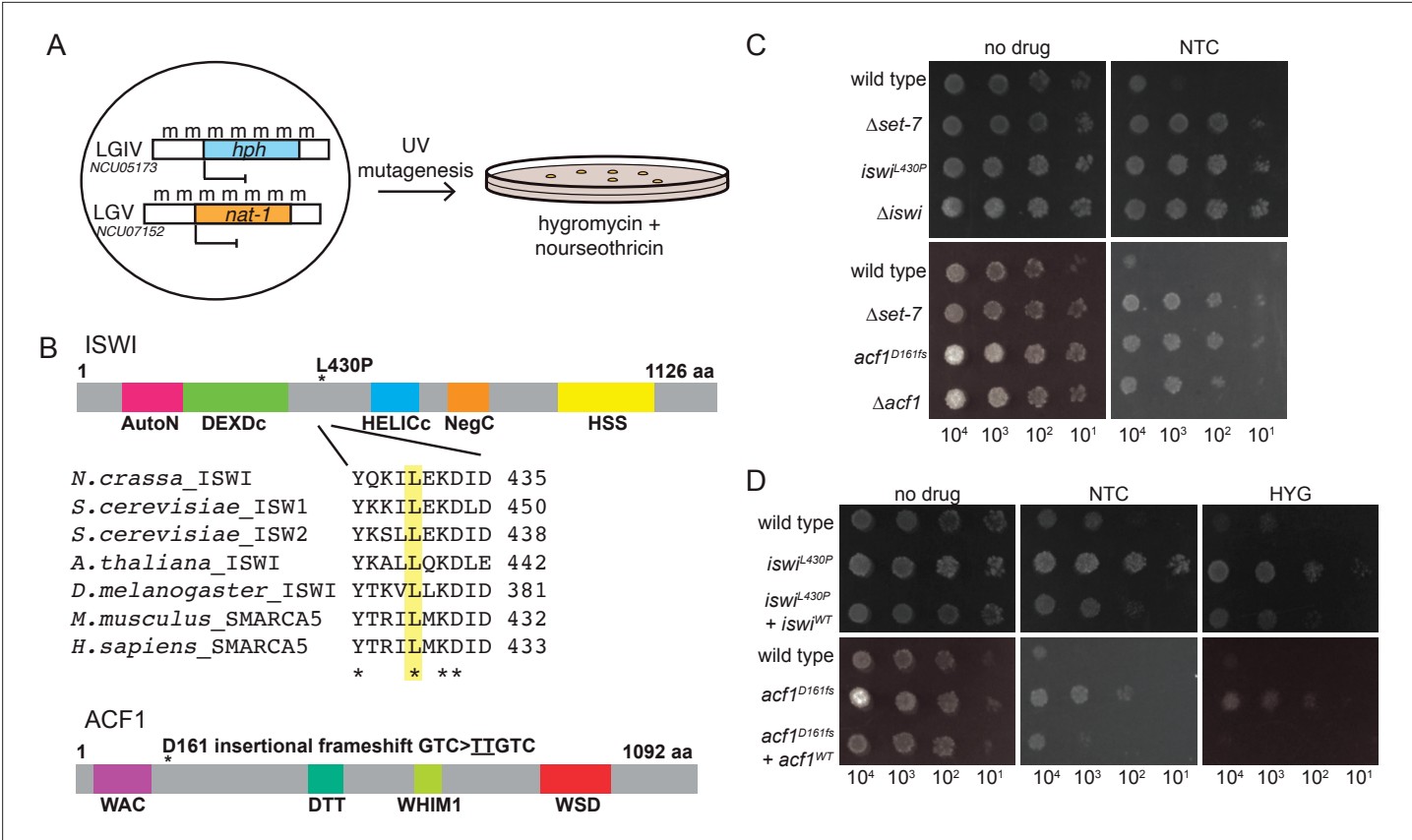

**Figure 1.** Forward genetics identifies ISWI complex members required for repression of H3K27-methylated genes. (**A**) Selection scheme with reporter genes inserted at H3K27 methylation-marked loci to select for genes required for silencing. (**B**) Schematic of protein domains in ISWI and ACF1 with the changes identified in our selection (L430P and D161fs, respectively; marked with asterisks). The conserved nature of the changed residue in ISWI is highlighted for the designated species. (**C**) Serial dilution spot-test silencing assay for the indicated strains, which all contain $P_{NCU07152}$::*nat-1* on media with or without nourseothricin (NTC). (**D**), Serial dilution spot-test silencing assay for the indicated strains, which contain $P_{NCU07152}$::*nat-1* and $P_{NCU5173}$::*hph*, on media with or without nourseothricin (NTC) or hygromycin (HYG). For complementation tests, wild-type copies of each gene were inserted at the *his-3* locus (indicated at left as +*iswi*^WT or +*acf1*^WT). All spot tests were imaged after 48 hr at 32°C and performed at least twice. The number of cells spotted is indicated beneath the images.

The online version of this article includes the following figure supplement(s) for figure 1:

**Figure supplement 1.** Genetic mapping and growth rate analysis of mutants identified in the selection.

## Results

### Forward genetic selection for genes required for Polycomb silencing identifies *iswi* and *acf1*

We previously designed and employed a forward genetic selection to identify novel genes required for silencing of H3K27-methylated genes (*McNaught et al., 2020*; *Wiles et al., 2020*). Briefly, the open reading frames of two genes (*NCU05173* and *NCU07152*) that require the H3K27 methyltransferase (SET-7) for repression were replaced with the hygromycin (*hph*) and nourseothricin (*nat-1*) resistance genes, respectively (*Figure 1A*). The resulting strain, which was sensitive to both antibiotics, was subjected to UV mutagenesis and grown on hygromycin- and nourseothricin-containing medium to select for drug-resistant colonies. As a step to identify mutations causing drug resistance, these strains, which were in an Oak Ridge genetic background, were crossed to the polymorphic Mauriceville strain (*Metzenberg et al., 1985*). Spores from these crosses were germinated on hygromycin- and/or nourseothricin-containing media to select for the mutant segregants, and genomic DNA from the progeny was pooled for whole-genome sequencing. Critical mutations were mapped by examining the ratio of Oak Ridge to Mauriceville single-nucleotide polymorphisms (SNPs) genome-wide

(*Pomraning et al., 2011*). Genetic variants were identified within the mapped regions using published tools (*Danecek et al., 2011*; *Garrison and Marth, 2012*).

SNP mapping for one mutant identified a region on linkage group VI that contained essentially 100% Oak Ridge SNPs, indicating the likely position of the critical mutation (*Figure 1—figure supplement 1A*). Within this region, we found a point mutation (CTT → CCT) predicted to cause a leucine to proline substitution at a conserved position in *iswi* (*NCU03875*) (L430P; *Figure 1B*). This same approach was used on a second mutant to map and identify a two base pair insertion (GTC → TTGTC) on linkage group III (*Figure 1—figure supplement 1B*) that leads to a frameshift (D161fs) in *acf1* (*NCU00164*) (*Figure 1B*).

To test if deletion of these two identified genes would also cause derepression, we created strains with the *NCU07152::nat-1* replacement and either Δ*iswi* or Δ*acf1* alleles. Indeed, deletion of either *iswi* or *acf1* resulted in nourseothricin-resistance, equivalent to the original mutants identified in our selection (*Figure 1C*). In addition, we showed that introduction of an ectopic, wild-type copy of *iswi* or *acf1* into the corresponding original mutant strain largely restored silencing of both H3K27 methylation mutant selection genes (*Figure 1D*). We noticed that disruption of *iswi* or *acf1* resulted in an early conidiation phenotype (production of asexual spores) that appeared as more dense growth in the spot tests (*Figure 1C and D*), but this was not accompanied by an increased linear growth rate. In fact, the Δ*iswi* strain showed a decreased growth rate relative to wild type; the Δ*acf1* strain grew comparably to wild type (*Figure 1—figure supplement 1C*). Taken together, these data confirm that *iswi* and *acf1* are required to maintain the drug sensitivity of strains containing the H3K27 methylation mutant selection genes and are thus good candidates for genes involved in repression of H3K27-methylated chromatin.

## ISWI and ACF1 form a complex in *N. crassa*

Evidence from several organisms, most notably budding yeast and *Drosophila*, has implicated ISWI as the catalytic subunit of several chromatin-remodeling complexes (*Petty and Pillus, 2013*). To look for possible ISWI-containing protein complexes in *N. crassa*, we affinity-purified overexpressed 3xFLAG-ISWI from *N. crassa* cellular extracts. Immunopurified samples were digested down to peptides and analyzed by mass spectrometry (MS) to identify potential interacting proteins. We focused on proteins whose counts comprised greater than 0.4% of the total spectrum counts. ISWI co-purified with ACF1 as well as with CRF4-3 (NCU02684), a homolog of Ioc4 and member of the *Saccharomyces cerevisiae* Isw1b complex (*Vary et al., 2003*). Top hits from the MS results also included NCU00412 and NCU09388, proteins not known from *S. cerevisiae* (using NCBI BLASTP *Altschul et al., 1990*; *Figure 2*, *Figure 2—figure supplement 1*). Nevertheless, we considered that these could be members of ISWI complexes based on the high number of unique peptides detected and the presence of a WHIM domain in NCU00412 and a PHD domain in NCU09388—domains that are present in *S. cerevisiae* Isw1 complex members, Ioc3 and Ioc2, respectively (*Vary et al., 2003*). NCU00412 and NCU09388 were also identified in a recent independent analysis of ISWI-interacting proteins in *N. crassa* and named ISWI accessory factors 1 and 2 (IAF-1 and IAF-2), respectively (*Kamei et al., 2021*). CRF4-3 was not previously identified as an ISWI-interacting partner, but for consistency, we will adopt the new nomenclature and refer to this protein as ISWI accessory factor 3 (IAF-3).

To confirm these interactions and to gain information on the possible formation of ISWI-containing subcomplexes, we engineered a C-terminal HA tag at the endogenous locus of each of the four most prominent putative ISWI-interacting partners: ACF1, IAF-3, IAF-1, and IAF-2. These proteins were purified by immunoprecipitation and subjected to MS to identify interacting partners. Interactions between ISWI and all four proteins were confirmed, with each HA-tagged protein pull-down yielding high unique peptide counts for ISWI. Additional interactions, with lower unique peptide counts, and typically lack of reciprocal pull-downs, were also found (*Figure 2A* and *Figure 2—figure supplement 1A*). These data suggest that ISWI forms multiple distinct protein complexes and, importantly, that ISWI and ACF1, two proteins identified in our selection for factors involved in the repression of H3K27-methylated genes, interact. The ACF1-HA pull-down identified two histone fold proteins, NCU03073 (HFP-1) and NCU06623 (HFP-2) (*Borkovich et al., 2004*; *Kamei et al., 2021*), as interacting partners (*Figure 2A* and *Figure 2—figure supplement 1A*). These proteins are notable because histone fold proteins are found in the CHRAC complex (DPB4 and DLS1 in *S. cerevisiae* and CHRAC14/16 in *D.*

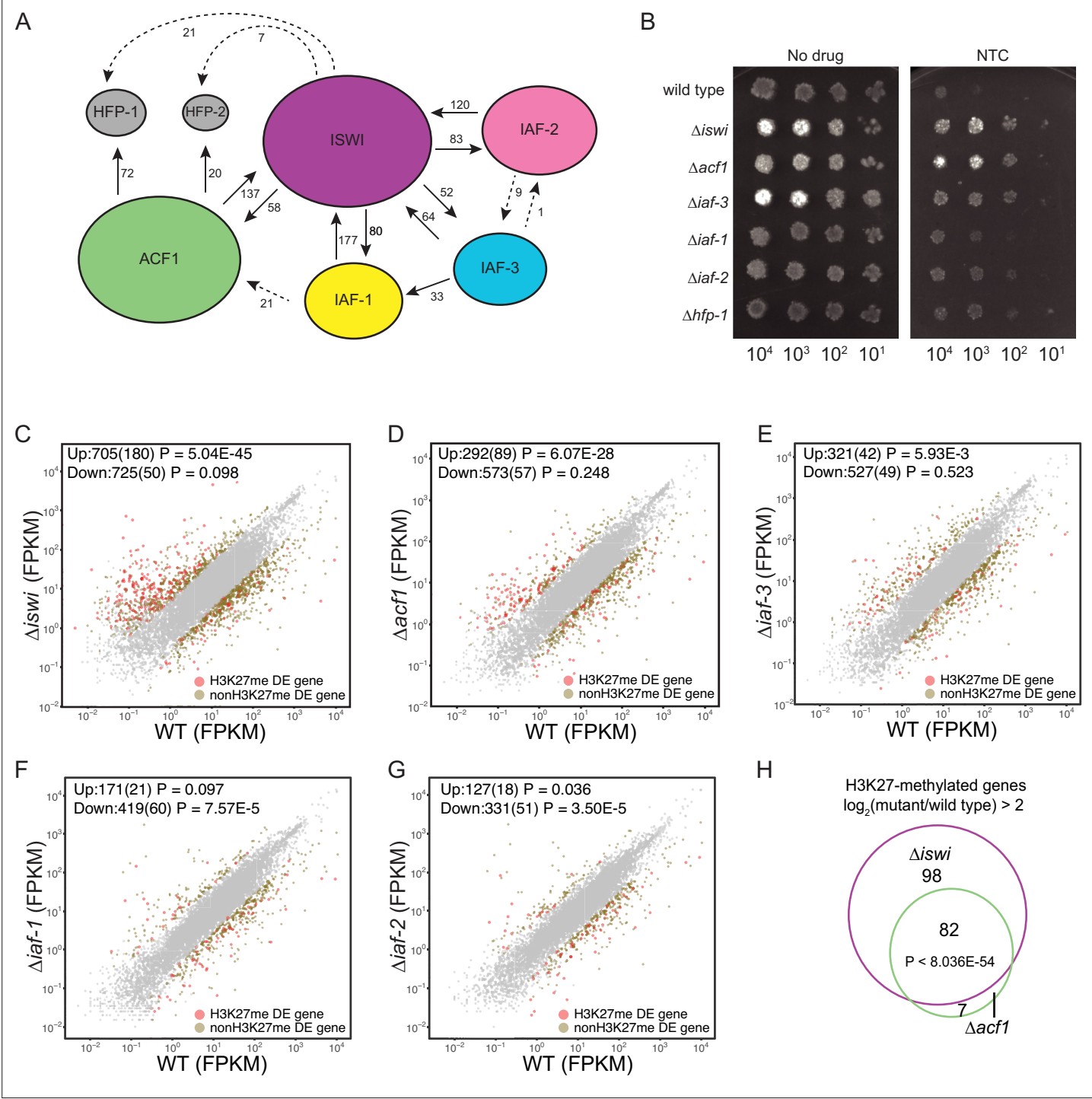

**Figure 2.** ISWI and ACF1 interact in *Neurospora crassa* and are required for repression of a subset of SET-7-repressed genes. (**A**) Schematic representation of ISWI-interactions found by immunoprecipitation followed by mass spectrometry. Proteins (ISWI/NCU03875, ACF1/NCU00164, IAF-3/NCU02684, IAF-1/NCU00412, IAF-2/NCU09388, HFP-1/NCU03073, and HFP-2/NCU06623) are depicted to scale. Arrows are drawn from the protein used as the 'bait' to the protein partner identified, and unique peptide counts are indicated. Dotted arrows indicate the peptide count was below 0.4% of the total spectrum threshold. Proteins in gray (HFP-1 and HFP-2) were identified as interacting partners but were not used as 'bait'. (**B**) Serial dilution spot-test silencing assay for the indicated strains on media with or without nourseothricin (NTC). All strains have $P_{NCU07152}::nat-1$. The number of cells spotted is indicated beneath the images, which were generated after incubation for 48 hr at 32°C. Spot test assays were repeated at least twice. (**C–G**) Expression level (FPKM) for each gene in the indicated mutant strain plotted against the expression level in wild type. Two biological replicates were used for each mutant. Two biological replicates were perfomed twice for wild type. Differentially expressed (DE) genes were defined using a significance

*Figure 2 continued on next page*

*Figure 2 continued*

cutoff of log₂fold change>2 for upregulated genes and log₂fold change<−2 for downregulated genes with a p value <0.05. Gray dots indicate genes that are not considered DE. Upper left corner shows the total number of significantly up- and downregulated genes with the number of H3K27-methylated genes in parentheses. Significance for enrichment of H3K27-methylated genes in each DE gene set was calculated by Fisher's exact test (FPKM - fragments per kilobase per million reads). (**H**) Venn diagram showing overlap between H3K27-methylated genes that are upregulated (log₂fold change>2; p value <0.05) in Δ*iswi* and Δ*acf1* strains. Significant overlap (p<8.036E−54) determined by hypergeometric probability test.

The online version of this article includes the following source data and figure supplement(s) for figure 2:

**Source data 1.** ISWI interactor comparison total spectra greater than 0.4 from mass spectrometry.

**Source data 2.** All mass spectrometry data.

**Source data 3.** mRNA-seq analysis.

**Figure supplement 1.** Summary of unique peptide counts from immunoprecipitation followed by mass spectrometry.

**Figure supplement 2.** *iswi* and *acf1* are required for regulation of non-H3K27-methylated genes.

*melanogaster*) along with ISWI and ACF (*Varga-Weisz et al., 1997*; *Corona et al., 2000*; *Iida and Araki, 2004*).

To investigate if any of the identified ISWI-interacting proteins, beyond ACF1, are involved in H3K27-methylated gene silencing, we first examined whether they are required for silencing the *NCU07152::nat-1* selection marker. As previously shown, deletion of *iswi* or *acf1* results in robust growth on nourseothricin, indicating strong derepression of the *nat-1* gene. Deletion of *iaf-3* or *hfp-1* also derepressed the *nat-1* marker. Strains with deletion of *iaf-1* and *iaf-2* showed more modest growth on nourseothricin (*Figure 2B*). These data show that ISWI, ACF1, and other ISWI-interacting proteins contribute to the silencing of the *NCU07152::nat-1* selection marker.

## *iswi* and *acf1* are required for repression of a subset of H3K27-methylated genes

We performed mRNA-seq on Δ*iswi* and Δ*acf1* strains to determine if the loss of these genes affects transcription beyond the *NCU07152::nat-1* selection marker and, if so, to determine if these effects were specific to H3K27-methylated domains, or were more general. We found that while the majority of gene expression changes observed upon loss of ISWI or ACF1 occurred outside of H3K27-methylated domains (*Figure 2—figure supplement 2A, B*), genes marked by H3K27 methylation were significantly enriched in the upregulated gene sets for Δ*iswi* and Δ*acf1* strains (*Figure 2C and D*). To determine the extent to which the other three ISWI-interacting partners contribute to silencing in H3K27-methylated regions, we performed mRNA-seq on strains with deletions of *iaf-3*, *iaf-1*, or *iaf-2*. We found that H3K27-methylated genes were modestly enriched in the Δ*iaf-3* and Δ*iaf-2* gene sets (*Figure 2E and F*) but not enriched in the Δ*iaf-1* gene set (*Figure 2G*). Nearly all (92%) of the H3K27-methylated genes that were upregulated in Δ*acf1* were also upregulated in Δ*iswi*, showing significant (p<8.036E−54) overlap between these two gene sets (*Figure 2H*). Only 30% of these genes were part of the Δ*set-7* upregulated gene set (*Figure 2—figure supplement 2C*), consistent with the notion that the repression of H3K27-methylated genes is not solely a result of PRC2 activity. This demonstrates that ISWI and ACF1 are not simply involved in the repression of the two H3K27-methylated genes that we used in our initial selection (*NCU05173* and *NCU07152*) but are also necessary for the repression of a large overlapping set of H3K27-methylated genes.

## *iswi* is required for wild-type H3K27 methylation and H3K36 trimethylation

We know that loss of H3K27 methylation (*Jamieson et al., 2013*) or loss of H3K36 methylation (*Bicocca et al., 2018*) is associated with derepression of genes in facultative heterochromatin. To investigate if upregulation of genes in facultative heterochromatin in Δ*iswi* and Δ*acf1* strains is due to loss of H3K27- or H3K36-methylation in these regions, we performed ChIP-seq for H3K27me2/3, H3K36me2, and H3K36me3. We compared the level of each of these histone modifications in Δ*iswi* and Δ*acf1* strains to that of wild type. We found that changes in H3K36me2 in each of these mutants were negligible (*Figure 3—figure supplement 1A, B*). We saw minor loss of both H3K27me2/3 and H3K36me3 in Δ*acf1*, and more changes in these histone marks in Δ*iswi* strains

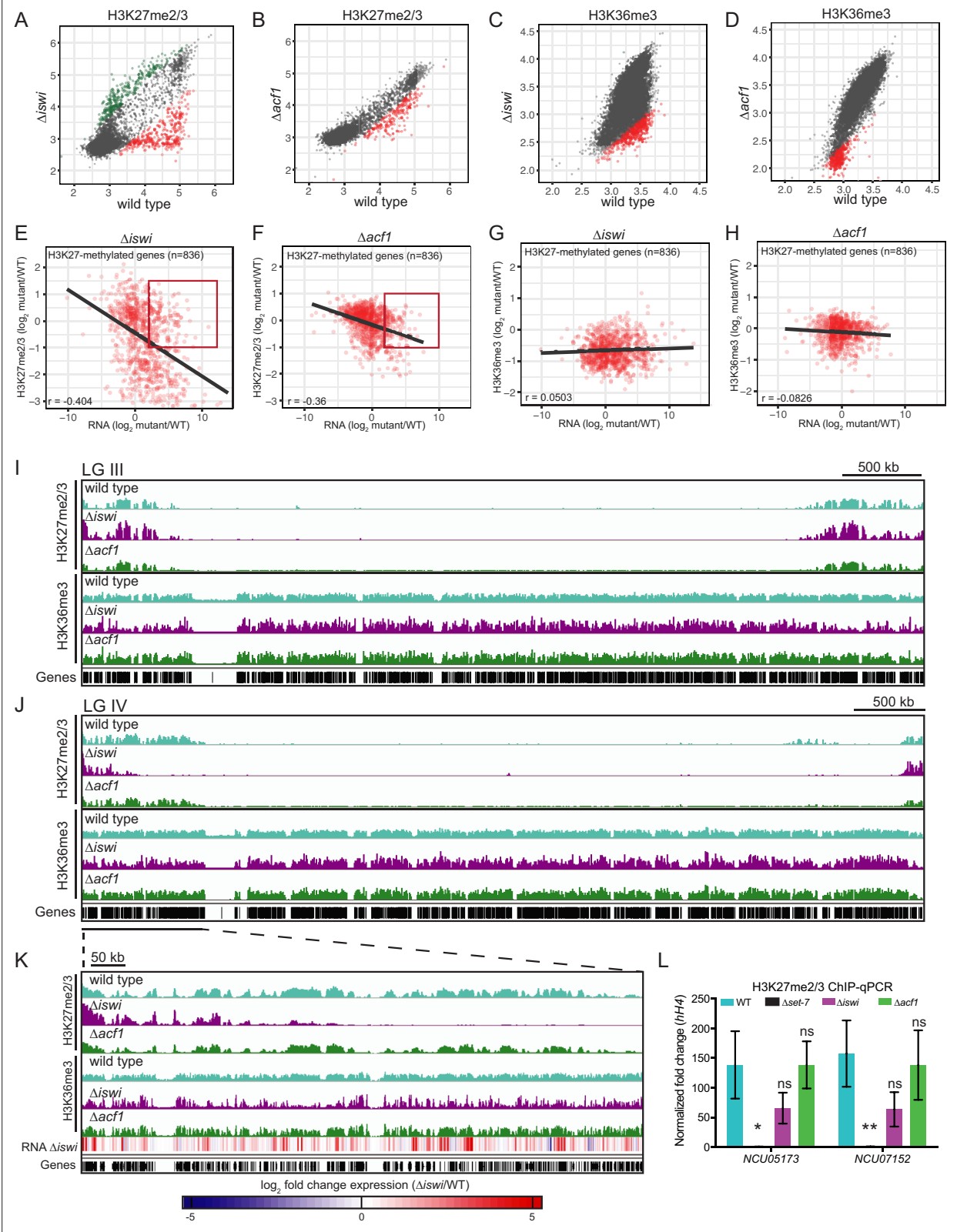

**Figure 3.** *iswi* and *acf1* are required for wild-type H3K27me2/3 and H3K36me3 but loss of these methyl marks is not required for transcriptional upregulation. (**A, B**) Scatter plots show the correlation of H3K27me2/3 at genes in wild type and Δ*iswi* or Δ*acf1* based on biological replicates of ChIP-seq data. Green points (n=260 in Δ*iswi* and n=0 in Δ*acf1*) represent genes with increased H3K27me2/3 levels (at least twofold over wild type) and red points (n=341 in Δ*iswi* and n=193 in Δ*acf1*) represent genes with decreased H3K27me2/3 levels (at least twofold relative to wild type) in the indicated

*Figure 3 continued on next page*

*Figure 3 continued*

mutant. (**C, D**) Scatter plots show the correlation of H3K36me3 at genes in wild type and Δ*iswi* or Δ*acf1* based on biological replicates of ChIP-seq data. Green points (n=1 in Δ*iswi* and n=0 in Δ*acf1*) represent genes with increased H3K36me3 levels (at least twofold over wild type) and red points (n=444 in Δ*iswi* and n=317 in Δ*acf1*) represent genes with decreased H3K36me3 levels (at least twofold relative to wild type) in the indicated mutant. (**E, F**) Scatter plots show the correlation between H3K27me2/3 and gene expression at H3K27-methylated genes (n=836) in the indicated mutants. Pearson correlation coefficient is reported. Red box indicates genes (n=92 in Δ*iswi* and n=66 in Δ*acf1*) that are significantly upregulated (log$_2$ fold change>2) but show no significant loss of H3K27me2/3 (log$_2$ fold change>–1). (**G, H**) Scatter plots show the correlation between H3K36me3 and gene expression at H3K27-methylated genes (n=836) in the indicated mutants. Pearson correlation coefficient is reported. (**I**) ChIP-seq tracks showing average level of H3K27me2/3 or H3K36me3 merged from two biological replicates for the indicated strains on LG III. Y-axis is 0–1000 RPKM for H3K27me2/3 tracks and 0–100 average read counts for H3K36me3 tracks. (**J**) Same as in (**I**), but for LG IV. (**K**) Enlarged ChIP-seq tracks showing the underlined region on LG IV from (**J**). Gene expression changes in Δ*iswi* are shown. (**L**) ChIP-qPCR data for H3K27me2/3 at the two genes used for the initial mutant selection (*NCU05173* and *NCU07152*) in the indicated strains. Filled bars represent the mean of technical triplicates and error bars show standard deviation (** for p<0.01, * for p<0.05, and ns for not significant; all relative to wild type by unpaired t-test). Data are from one representative experiment that was performed three times.

The online version of this article includes the following source data and figure supplement(s) for figure 3:

**Source data 1.** H3K27me2/3 ChIP-seq comparisons (Δ*iswi* and Δ*acf1*).

**Source data 2.** H3K36me3 ChIP-seq comparisons.

**Source data 3.** Comparison of H3K27me2/3 ChIP-seq data and RNA-seq data in Δ*iswi* and Δ*acf1*.

**Figure supplement 1.** *iswi* and *acf1* are not required for H3K36me2.

**Figure supplement 1—source data 1.** H3K36me2 ChIP-seq comparisons.

**Figure supplement 2.** Loss of *iaf-3*, *iaf-1*, and *iaf-2* results in minor changes in H3K27me2/3.

**Figure supplement 2—source data 1.** H3K27me2/3 ChIP-seq comparisons (Δ*iswi*, Δ*acf1*, Δ*iaf-3*, Δ*iaf-1*, and Δ*iaf-2*).

(***Figure 3A–D***). H3K27me2/3 ChIP-seq of strains with genes for other ISWI-interacting proteins deleted (Δ*iaf-3*, Δ*iaf-1*, or Δ*iaf-2*) showed only minor changes in H3K27me2/3 (***Figure 3—figure supplement 2A-C***).

## Loss of H3K27me2/3 or H3K36me3 in Δ*iswi* and Δ*acf1* strains is not required for transcriptional upregulation of genes in facultative heterochromatin

We next asked if these changes in H3K27 or H3K36 methylation correlated with changes in gene expression in H3K27-methylated regions. We found that there was a negative correlation between H3K27me2/3 and gene expression in both Δ*iswi* (r=–0.404) and Δ*acf1* (r=–0.36) strains (***Figure 3E and F***) while there was no correlation between H3K36me3 and gene expression in these strains (***Figure 3G and H***). Despite the negative correlation between H3K27me2/3 level and gene expression, the majority of upregulated genes—51% and 74% for Δ*iswi* and Δ*acf1*, respectively—had no significant loss of H3K27 methylation (***Figure 3E and F***). We looked at the distribution of H3K27me2/3 and H3K36me3 along the chromosomes (***Figure 3I and J*** and ***Figure 3—figure supplement 2D***) and found that while much of the H3K27me2/3 and H3K36me3 resembled wild type, large domains of H3K27me2/3 were lost in Δ*iswi* strains whereas more discrete decreases in H3K36me3 were observed in Δ*iswi* strains and to a lesser degree in Δ*acf1* strains (***Figure 3I–K***).

We examined the gene expression changes along the left arm of LG IV in the Δ*iswi* strain and confirmed that many upregulated genes fell in regions that showed wild-type H3K27me2/3 and H3K36me3 (***Figure 3K***). ChIP-qPCR at the H3K27-methylated marker genes (*NCU05173* and *NCU07152*) further validated the finding that loss of H3K27 methylation is not required for transcriptional upregulation in Δ*iswi* and Δ*acf1* strains (***Figure 3L***). This is consistent with our previous findings showing that loss of H3K27 methylation is not a prerequisite for upregulating genes in facultative heterochromatin (***Wiles et al., 2020***). Taken together, these data show that *iswi* is required for normal H3K27me2/3 and H3K36me3, while loss of *acf1* results in minor changes, suggesting that the ACF complex does not play a major role in directing or maintaining these histone modifications. Furthermore, loss of these histone marks is not required for transcriptional upregulation in facultative heterochromatin.

## SET-7 promotes ACF1 association with facultative heterochromatin

To identify chromatin targets of the *N. crassa* ACF complex, we fused the *Escherichia coli* DNA adenine methyltransferase (*van Steensel and Henikoff, 2000*) to the C-terminus of endogenous ACF1 and assayed adenine methylated DNA fragments by sequencing (DamID-seq) (*Zhou, 2012*). We found that ACF1 localization is not restricted to one part of the genome, but rather appears to interact with chromatin genome-wide (*Figure 4A and B*). However, when *set-7* was deleted, eliminating H3K27 methylation, ACF1 localization to H3K27-methylated regions was reduced relative to wild type, suggesting that H3K27 methylation, or SET-7 presence, promotes ACF1 interactions specifically with these genomic regions (*Figure 4A and B*, *Figure 4—figure supplement 1A, B*). These results were confirmed for two H3K27 methylation-marked regions (*NCU05173* and Tel VIIL) by Southern hybridizations with genomic DNA from DamID experiments. In contrast, deletion of *set-7* had no effect on ACF1-Dam localization at a euchromatic region (*his-3*) (*Figure 4—figure supplement 1A*). When we compared ACF1-Dam localization to that of a nonspecific control (Dam only; referred to as Free-Dam), we found that both constructs localized to non-H3K27-methylated genes at similar levels, and this was independent of *set-7* presence (*Figure 4C*). In contrast, ACF1-Dam localized to H3K27-methylated genes more than Free-Dam and this increased localization was partially dependent on *set-7* (*Figure 4D*). These data suggest that ACF1 association with facultative heterochromatin is promoted by, but not fully dependent on, an intact PRC2 complex and/or H3K27 methylation.

## Loss of ACF has minor effects on nucleosome spacing

ACF-like complexes function differently in flies and yeast. In *D. melanogaster*, ACF acts globally to space nucleosomes evenly (*Baldi et al., 2018*), whereas in *S. cerevisiae*, the analogous Isw2 complex specifically moves the +1 nucleosome in the 5' direction, toward the nucleosome-depleted region (NDR) (*Whitehouse et al., 2007*; *Yen et al., 2012*). To characterize nucleosome positioning in wild-type and mutant strains of *N. crassa*, we performed MNase digestion followed by high-throughput sequencing (MNase-seq). We first looked at nucleosome repeat length using the autocorrelation function (*Braunschweig et al., 2009*), which can analyze nucleosome positions independent of the TSS. When we looked genome-wide or considered only H3K27-methylated regions, we found only minor changes in nucleosome repeat length between wild-type and mutant strains (Δ*iswi*, Δ*acf1*, Δ*iaf-3*, Δ*iaf-1*, Δ*iaf-2*, and Δ*set-7*) (*Figure 5—figure supplement 1A, B*). This suggested that ISWI-containing complexes do not have major contributions to global nucleosome spacing in *N. crassa* or there is redundancy among these proteins.

## Loss of ACF results in a downstream shift of the +1 nucleosome and transcriptional upregulation at a subset of H3K27-methylated genes

We next considered that the *N. crassa* ACF complex may function more like the *S. cerevisiae* Isw2 complex. For this analysis, we looked at nucleosome positions in the promoter region of genes that had regular nucleosome arrays (defined as spectral density SD; *Baldi et al., 2018* score>2; n=7753) in at least one strain (wild type, Δ*iswi*, Δ*acf1*, Δ*iaf-3*, Δ*iaf-1*, Δ*iaf-2*, or Δ*set-7*). We found that when all SD genes were considered, deletion of *iswi* or *acf1* was more likely to result in a downstream shift (>30 bp) of the +1 nucleosome than when *iaf-3*, *iaf-1*, *iaf-2*, or *set-7* were deleted (*Figure 5—figure supplement 2A*). This trend held when only H3K27-methylated SD genes (n=358) were considered (*Figure 5A*). Importantly, a significant portion of the H3K27-methylated genes with a shifted nucleosome is shared between *iswi* and *acf1* (p<9.91E−13) (*Figure 5B*). These data suggest that ISWI and ACF1 may work in concert to position the +1 nucleosome at a subset of genes, including those in H3K27-methylated regions.

Analysis of nucleosome positions at all SD genes in wild-type and mutant strains (Δ*iswi*, Δ*acf1*, Δ*iaf-3*, Δ*iaf-1*, Δ*iaf-2*, and Δ*set-7*) revealed some differences in occupancy at the −1 nucleosome but no global shift in nucleosome positions (*Figure 5—figure supplement 2B*). Because a genome-wide view can mask changes at specific targets and because the characteristic Isw2 5' 'pulling' activity can only be appreciated when a subset of targets are examined (*Yen et al., 2012*; *Ocampo et al., 2016*; *Donovan et al., 2021*), we sought to limit our analysis to genes that might be targets of the ACF complex. Our inability to perform chromatin immunoprecipitation (ChIP) on ACF1 and limitations of DamID-seq precluded a strict analysis of direct ACF1 targets. Considering that our data support a functional role in transcriptional repression at H3K27-methylated genes, we restricted our analysis

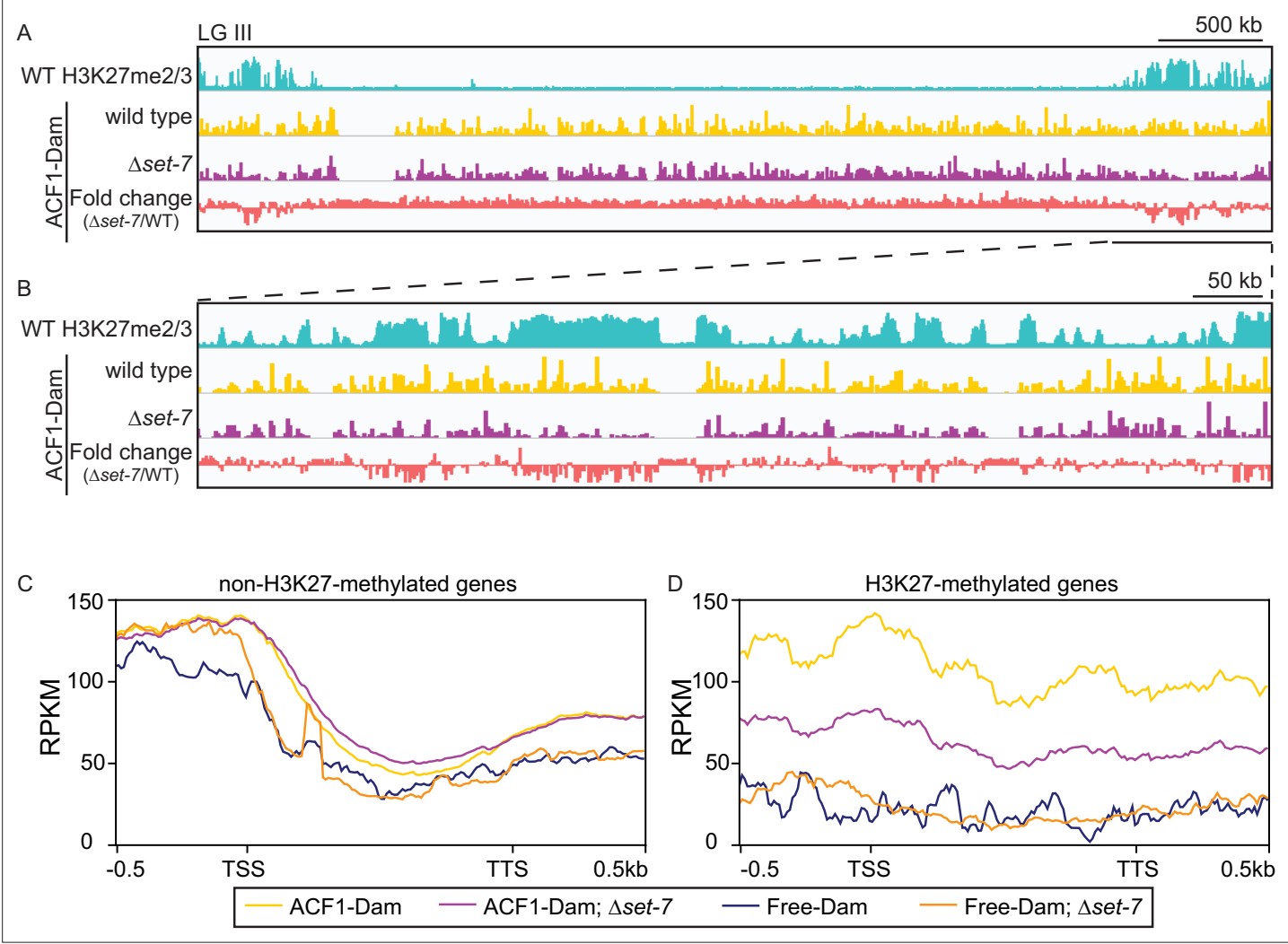

**Figure 4.** ACF1 localizes to H3K27me2/3-marked regions of the genome. (**A**) Top track shows wild-type H3K27me2/3 levels based on ChIP-seq averaged from two biological replicates for one chromosome (LG III). Y-axis is 0–500 RPKM. Middle two tracks show DamID-seq average reads merged from two biological replicates for the indicated genotypes. Y-axis is 0–500 RPKM. Bottom track compares the DamID-seq reads from Δ*set-7* strains to wild-type strains (shown above) displayed as the fold change between the two genotypes. Y-axis is –3–3. (**B**) Same as in (**A**), but showing an enlarged view of the right arm of LG III. Region shown is underlined in black in (**A**). (**C**) Average enrichment based on DamID-seq for each non-H3K27-methylated gene, scaled to 1 kb, ±500 base pairs, is plotted for the indicated strains. All lines represent average reads from two biological replicates except for Free-Dam which is from only one. TSS, transcription start site; TTS, transcription termination site. (**D**) Same as in (**C**), but for H3K27-methylated genes.

The online version of this article includes the following source data and figure supplement(s) for figure 4:

**Figure supplement 1.** ACF1 localizes to H3K27me2/3-marked regions of the genome.

**Figure supplement 1—source data 1.** Raw image for Et-Br gel.

**Figure supplement 1—source data 2.** Raw image for Southern blot probed with *NCU05173.*

**Figure supplement 1—source data 3.** Raw image for Southern blot probed with Tel VIIL.

**Figure supplement 1—source data 4.** Raw image for Southern blot probed with *his-3.*

**Figure supplement 1—source data 5.** Raw, uncropped image for Et-Br gel with labels.

**Figure supplement 1—source data 6.** Raw, uncropped image for Southern blot probed with *NCU05173* with labels.

**Figure supplement 1—source data 7.** Raw, uncropped image for Southern blot probed with Tel VIIL with labels.

**Figure supplement 1—source data 8.** Raw, uncropped image for Southern blot probed with *his-3* with labels.

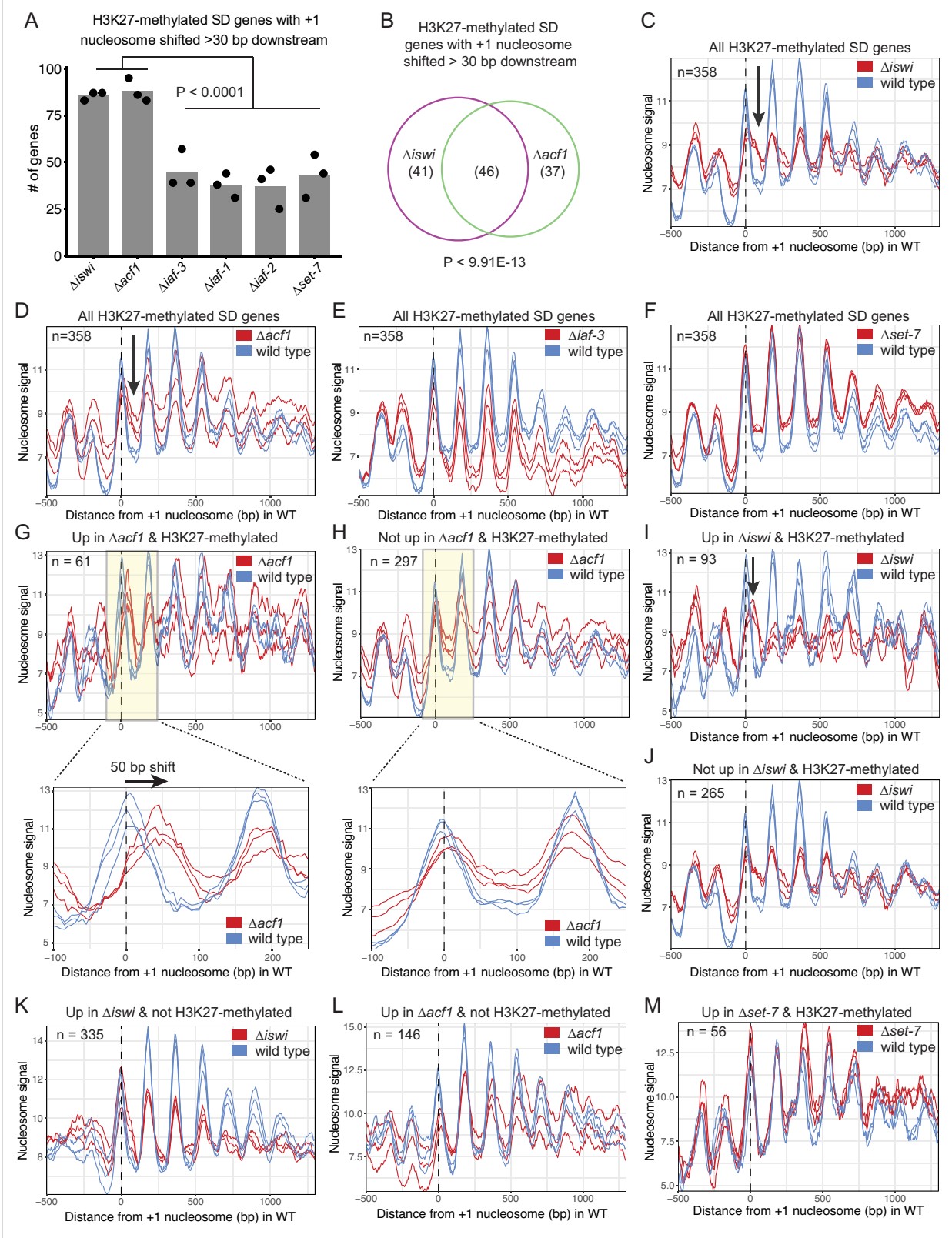

**Figure 5.** ISWI and ACF1 position the +1 nucleosome at H3K27-methylated, upregulated genes. (**A**) Histogram of the number of H3K27-methylated SD genes (spectral density score for nucleosome order>2; n=358) that have the +1 nucleosome shifted downstream >30 base pairs when compared to wild type in the indicated mutant strains. Each point represents biological replicate 1, biological replicate 2, or analysis of the merged replicates and filled bar is the average of all three values. P values were determined with an unpaired t-test. (**B**) Venn diagram showing overlap of H3K27-methylated

*Figure 5 continued*

SD genes with a +1 nucleosome shifted downstream >30 bp when *iswi* or *acf1* is deleted. P value was determined by hypergeometric probability test. (C–F) Average nucleosome signal at all H3K27-methylated SD genes plotted from MNase-seq data for the indicated mutants and wild type. The three colored lines represent biological replicate 1, biological replicate 2, and the average of the replicates for the strains indicated in the key. Arrows in (C) and (D) indicate the shifted +1 nucleosome. (G) Average nucleosome signal at SD genes that are upregulated (FDR <0.05) and marked by H3K27 methylation in Δ*acf1* strains. The three colored lines represent biological replicate 1, biological replicate 2, and the average of the replicates. The boxed, shaded region is enlarged in the lower panel. (H) Same as panel (G), but for H3K27-methylated SD genes that are not upregulated in Δ*acf1* strains. (I) Average nucleosome signal at SD genes that are upregulated (FDR <0.05) and marked by H3K27 methylation in Δ*iswi* strains. The three colored lines represent biological replicate 1, biological replicate 2, and the average of the replicates. Arrow indicates the shifted +1 nucleosome. (J) Same as (I), but for H3K27-methylated SD genes that are not upregulated in Δ*iswi* strains. (K, L) Average nucleosome signal at SD genes that are upregulated (FDR <0.05) and not marked by H3K27 methylation in Δ*iswi* (K) and Δ*acf1* (L) strains. The three colored lines represent biological replicate 1, biological replicate 2, and the average of the replicates. (M) Same as (I), but for H3K27-methylated SD genes that are upregulated in Δ*set-7*.

The online version of this article includes the following source data and figure supplement(s) for figure 5:

**Source data 1.** List of SD genes used for MNase-seq analysis.

**Figure supplement 1.** ISWI and its interacting partners have minor effects on nucleosome repeat length in *Neurospora crassa*.

**Figure supplement 2.** Nucleosome shifts are specific to genes that are H3K27-methylated and upregulated in Δ*iswi* and Δ*acf1*.

to these regions (H3K27-methylated SD genes; n=358). The MNase signal plots revealed that the +1 nucleosome shifted downstream in the absence of *iswi* or *acf1* (*Figure 5C and D*); in contrast, no shift was seen when other ISWI-interacting partners (Δ*iaf-3*, Δ*iaf-1*, and Δ*iaf-2*) were deleted (*Figure 5E*; *Figure 5—figure supplement 2C, D*). There was also no shift observed in Δ*set-7* strains when all H3K27-methylated SD genes were considered (*Figure 5F*). These findings suggest that ISWI and ACF1 act to position the +1 nucleosome at a substantial subset of H3K27-methylated genes. Furthermore, SET-7, and hence H3K27 methylation, is not required for nucleosome positioning by ISWI/ACF1.

To test if the downstream nucleosome shift at H3K27-methylated genes in Δ*iswi* or Δ*acf1* strains correlated with increased gene expression, we further focused our analysis to look at nucleosome positions in H3K27-methylated SD genes that were upregulated when *iswi* or *acf1* was deleted. We found that the +1 nucleosome shifted 50-bp downstream on average at H3K27-methylated SD genes that were upregulated (FDR <0.05) in Δ*acf1* strains (*Figure 5G*), whereas no such shift was seen in the +1 nucleosome of H3K27-methylated SD genes that were not upregulated in Δ*acf1* strains (*Figure 5H*). Similarly, H3K27-methylated genes that were upregulated (FDR <0.05) in Δ*iswi* display a more prominent downstream shift of the +1 nucleosome than those genes that were not upregulated (*Figure 5I and J*). Taken together, these data suggest that positioning of the +1 nucleosome by ISWI and ACF1 at a subset of H3K27-methylated genes contributes to transcriptional repression.

To ensure that the nucleosome shifts observed at H3K27-methylated, upregulated genes in Δ*iswi* or Δ*acf1* strains were not simply a consequence of the transcriptional activity, we looked at nucleosome positions in non-H3K27-methylated genes that are upregulated in these strains. We found no nucleosome shift at non-H3K27-methylated genes that are upregulated in Δ*iswi* or Δ*acf1* (*Figure 5K and L*). This suggests that the upregulation at non-H3K27-methylated targets is through a different mechanism or is an indirect effect. We also looked at the nucleosome positions in H3K27-methylated genes that are upregulated in Δ*set-7* (*Figure 5M*). We found no nucleosome shift at these genes. Taken together, this shows that transcriptional upregulation is not sufficient to induce a nucleosome shift. These findings support a model in which ACF acts directly at H3K27-methylated, upregulated genes.

## ACF is a new player in the multifaceted repression of facultative heterochromatin

To gain a better understanding of how histone remodeling by ACF fits into our current framework of transcriptional repression by facultative heterochromatin, we compared gene expression profiles for Δ*iswi* and Δ*acf1* to data sets for other genes that we identified as players in this repression: SET-7, the H3K27 methyltransferase; ASH1, an H3K36 methyltransferase; and EPR-1, an apparent H3K27 methyl-binding protein. We created a clustered heatmap of the gene expression data and found that five clusters emerged (*Figure 6A*). Cluster 1 included genes that were upregulated in every mutant strain except Δ*epr-1*. Cluster 2 contained genes that were highly upregulated in Δ*iswi*, *ash1*[Y888F] (a catalytic null) or Δ*acf1* strains, while cluster 3 genes were only upregulated in Δ*iswi* or *ash1*[Y888F] strains.

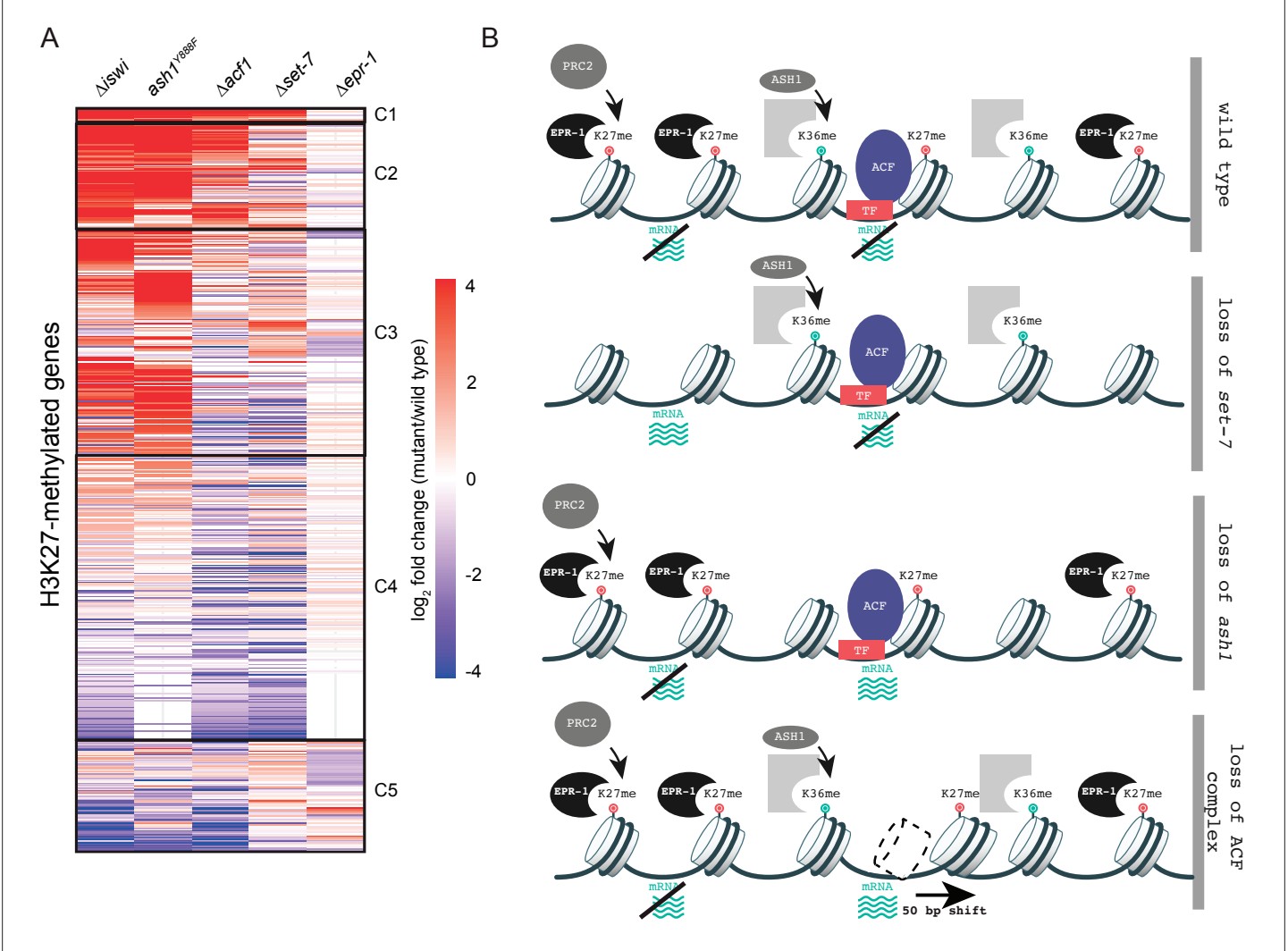

**Figure 6.** Multifaceted repression in facultative heterochromatin. (**A**) Clustered heatmap made using mRNA-seq data for combined biological replicates of the indicated mutant strains. All H3K27-methylated genes that had reads in mRNA-seq data were included (n=821). Clusters (C1–C5) were determined by eye. (**B**) Model depicting our current framework of factors responsible for maintaining gene silencing in regions marked by H3K27 methylation. Loss of this methyl-mark itself is sufficient to activate a fraction of genes, in part because of loss of the H3K27 methyl-specific factor EPR-1. Repression of many other genes, in H3K27-methylated domains and elsewhere, depend on both H3K36 methylation by ASH1 and both components of the ACF complex (ISWI and ACF). Gray partial square represents an unknown H3K36 methyl binding protein. TF represents unknown transcription factor(s) that could recruit/direct the ACF complex.

The online version of this article includes the following source data and figure supplement(s) for figure 6:

**Source data 1.** Heatmap clustering of mRNA-seq data.

**Figure supplement 1.** Loss of *ash1* function does not result in a downstream nucleosome shift.

The fourth and largest cluster contained genes that were not highly upregulated in any mutant strain. The final cluster contained genes with more varied expression among the mutant strains, but perhaps showed some enrichment for genes upregulated in Δ*set-7* and Δ*epr-1* strains.

The heatmap revealed striking overlap between gene expression profiles of Δ*iswi* and *ash1*^Y888F^ strains, which prompted us to investigate this relationship further. We previously showed that *iswi* is not required for H3K36me2 (*Figure 3—figure supplement 1*), the predominant histone mark catalyzed by ASH1. We explored the possibility that *ash1* is required for proper nucleosome positioning by the ACF complex. We performed MNase-seq in *ash1* mutant strains and looked at nucleosome positions at H3K27-methylated genes that are upregulated when *iswi* is deleted. In contrast to the prominent downstream nucleosome shift we saw in this gene set in Δ*iswi* strains, we observed no nucleosome

shift in *ash1* mutant strains (**Figure 6—figure supplement 1A**). These data suggest that ISWI and ASH1 act in distinct, perhaps parallel, pathways to regulate a common set of H3K27-methylated genes.

## Discussion

### Control and function of facultative heterochromatin reflect the superimposition of a constellation of molecular mechanisms

Pioneering work on the Polycomb system of *Drosophila* revealed that methylation of lysine 27 of histone H3, catalyzed by Enhancer of Zeste in the PRC2 complex, is associated with, and important for, gene repression in facultative heterochromatin (**Kassis et al., 2017**). Although much has been learned, the importance of Polycomb repression in the development of multicellular organisms has stymied progress toward a full understanding of its control and function. Moreover, there are indications of variable underlying mechanisms. For example, the PRC1 complex, which is widely regarded as central to Polycomb function in *Drosophila* and higher organisms, is less conserved than PRC2, and at least in some organisms, is absent (**Schuettengruber et al., 2017**). Similarly, Polycomb response elements (PREs), cis-acting DNA sequences controlling the distribution of H3K27 methylation in *Drosophila*, do not appear to be universal (**Kassis and Brown, 2013**). The complexity and importance of the Polycomb system in multicellular organisms led us to dissect the control and function of H3K27 methylation in the filamentous fungus *N. crassa*. We defined the PRC2 complex of Neurospora, demonstrated that it methylates H3K27 in roughly 7% of the genome, and is necessary for repression of scores of genes even though it is not essential in this organism (**Jamieson et al., 2013**; **McNaught et al., 2020**). Utilization of special genetic resources for Neurospora revealed that the organism has at least two distinct forms of H3K27 methylation (**Jamieson et al., 2018**), namely: 1. position-dependent, associated with telomere regions and characterized by involvement of the Neurospora p55 homolog (NPF) and the PRC2 accessory subunit (PAS) (**McNaught et al., 2020**), and 2. position-independent, which is found interstitially and does not depend on NPF or PAS (**Jamieson et al., 2018**). We have also previously shown that ASH1, an H3K36 methyltransferase, is critical for maintaining repression of many genes, including most of those in facultative heterochromatin (**Bicocca et al., 2018**).

   The nonessential nature of H3K27 methylation, and the convenience of Neurospora for genetics and biochemistry, allowed us to design and implement a powerful selection for mutants defective in silencing genes in facultative heterochromatin. This unbiased scheme revealed both expected factors required for repression, including members of the PRC2 complex, and unanticipated players such as EPR-1 (**Wiles et al., 2020**) and the ACF complex reported here. A particularly interesting general finding is that repression is not simply due to a linear pathway of factors. While some factors cooperate to maintain repression, results of mRNA-seq revealed considerable variation in the spheres of influence of the various factors. For example, loss of the H3K27 methyl-mark itself, or of the apparent H3K27 methyl-reader, EPR-1, each lead to derepression of somewhat different subsets of H3K27-methylated genes (**Wiles et al., 2020**), while loss of elements of the ACF remodeling machine leads to loss of silencing of a larger set of H3K27 methyl-marked genes, even without loss of this characteristic mark of facultative heterochromatin. The overall picture that is emerging is cartooned in **Figure 6B** with more specifics discussed below.

### The *N. crassa* ACF complex is required for transcriptional repression at facultative heterochromatin

Our genetic identification of *iswi* and *acf1* as genes required for silencing H3K27 methyl-marked loci is consistent with a growing body of evidence that chromatin structure plays a major role in the transcriptional status of genes. NDRs are characteristic of transcriptionally active promoters and are thought to allow access of the transcriptional machinery (**Lai and Pugh, 2017**). Conversely, nucleosomes can be positioned onto regulatory sequences in promoter regions by chromatin remodelers to cause repression (**Whitehouse and Tsukiyama, 2006**; **Whitehouse et al., 2007**). We found that nearly all H3K27-methylated genes that are upregulated in Δ*acf1* also showed increased expression in Δ*iswi*, whereas Δ*iswi* had several uniquely upregulated genes. This is consistent with a model in which ACF1 is required for targeting ACF to chromatin targets but requires ISWI to catalyze nucleosome

movement and allow for increased transcription. ISWI is also part of other protein complexes which may lead to direct or indirect upregulation of distinct genes.

ACF-like complexes are conserved from budding yeast (*Tsukiyama et al., 1999*) to humans (*LeRoy et al., 2000*), but most of the biochemical studies of these complexes have been done with yeast and flies, which, curiously, revealed apparent functional discrepancies. In yeast, Isw2 (homologous to the ACF complex of *Drosophila*) acts in promoter regions where it binds to the +1 nucleosome and moves it in the 5′ direction toward the NDR (*Whitehouse et al., 2007*; *Yen et al., 2012*; *Kubik et al., 2019*). In contrast, ACF in *Drosophila* has been characterized as a nonspecific nucleosome spacing and assembly factor promoting global chromatin regularity (*Baldi et al., 2018*). The distinct modes of action of ACF-like chromatin remodelers in yeast and *Drosophila* warrant further study in other organisms. Our investigation of nucleosome positioning activities of ISWI and ACF1 in *N. crassa* revealed that these factors are required for positioning the +1 nucleosome at a subset of genes, particularly those marked by H3K27 methylation. Thus, *N. crassa* ACF seems to function more like the *S. cerevisiae* Isw2 than the *D. melanogaster* ACF.

Although the detailed mechanism of recruitment and target selection for ACF-like complexes remains unclear, work in yeast implicates interactions of such complexes with transcription factors (*Goldmark et al., 2000*; *Donovan et al., 2021*). It was recently shown that the WAC domain of Itc1 in the Isw2 complex contains acidic residues required for binding to transcription factors and for nucleosome positioning at target promoters (*Donovan et al., 2021*). These residues (E33 and E40) are conserved in Neurospora (E32 and E39) but not in *Drosophila*, potentially accounting for the apparent less specific function of ACF in flies (*Donovan et al., 2021*). It will be of interest to determine if there are transcription factors that bind to facultative heterochromatin in *N. crassa* and mediate interactions with ACF1 to facilitate localization and activity of the ACF complex.

Our DamID-seq results are compatible with a 'continuous sampling' model proposed for some ISWI chromatin remodelers (*Erdel et al., 2010*). In this model, the ACF complex transiently interacts with chromatin (*Gelbart et al., 2005*) throughout the nucleus in an autoinhibited conformation (*Clapier and Cairns, 2012*; *Ludwigsen et al., 2017*) until some, still undefined, feature (*Clapier and Cairns, 2012*; *Hwang et al., 2014*; *Ludwigsen et al., 2017*; *Donovan et al., 2021*) releases the autoinhibition and allows it to engage, activate, and move nucleosomes by hydrolyzing ATP. Our results suggest that ACF localizes broadly throughout the genome but has specific activity at H3K27-methylated regions, raising the possibility that a feature of facultative heterochromatin influences ACF activity. The transient nature of this chromatin interaction could account for our inability to confirm our ACF1 DamID findings with ChIP. Attempts to identify the targets of the homologous complex by ChIP have been also unsuccessful in *Drosophila* (*Scacchetti et al., 2018*).

## The *N. crassa* ACF complex positions the +1 nucleosome in promoters of H3K27-methylated genes to mediate transcriptional repression

In theory, nucleosome movement could be either a cause or consequence of transcriptional activation. Our finding that H3K27-methylated genes did not show changes in the position of the +1 nucleosome when they are derepressed by deletion of *set-7* suggests that changes in nucleosome position are not simply due to transcriptional activation. Moreover, the fact that genes that are upregulated outside of facultative heterochromatin domains in strains with deletions of *iswi* or *acf1* do not display a nucleosome shift suggests that ACF acts directly and has some specificity for H3K27-methylated regions. These findings support the idea that transcriptional derepression of H3K27-methylated genes in Δ*iswi* and Δ*acf1* strains is a consequence of a misplaced +1 nucleosome. It is noteworthy that while Isw2-mediated repression is thought to occur by the placement of the +1 nucleosome over important DNA regulatory elements, occluding transcriptional machinery and/or general regulatory factors (*Whitehouse et al., 2007*; *Yen et al., 2012*), full repression at some targets, such as the early meiotic genes, also requires histone deacetylase activity from Rpd3 (*Goldmark et al., 2000*; *Fazzio et al., 2001*). Clearly, the mechanism of repression by ACF, including the identification of additional players, perhaps including transcription factors, histone deacetylases, and other chromatin modifying factors, deserves further study.

## Conclusions

Despite differences in the modes of action of *S. cerevisiae* Isw2 and *Drosophila* ACF, their biological outcomes are the same—transcriptional repression (*Goldmark et al., 2000*; *Fyodorov et al., 2004*; *Ocampo et al., 2016*; *Scacchetti et al., 2018*). We found that nucleosome positioning by the *N. crassa* ACF complex also leads to transcriptional repression, particularly at H3K27-methylated regions of the genome, establishing the ACF complex as a player in transcriptional repression characteristic of facultative heterochromatin. It will be valuable to determine if interplay between Polycomb-mediated repression and ISWI chromatin remodelers holds in other organisms. Interestingly, ACF has been indirectly linked to Polycomb repression in flies (*Scacchetti et al., 2018*), and notably, ISWI components were identified in a screen for factors required for Polycomb repression in mammalian cells (*Nishioka et al., 2018*), raising the possibility that the role of the ACF complex in Neurospora is general.

# Materials and methods

### Key resources table

| Reagent type (species) or resource | Designation | Source or reference | Identifiers | Additional information |
|---|---|---|---|---|
| Strain, strain background (*Neurospora crassa*) | Mauriceville | FGSC 2225 | N51 | *mat A*; Mauriceville |
| Strain, strain background (*N. crassa*) | Wild type | FGSC 2489 | N3752 | *mat A*; Oak Ridge |
| Strain, strain background (*N. crassa*) | Wild type | FGSC 4200 | N3753 | *mat a*; Oak Ridge |
| Strain, strain background (*N. crassa*) | *Sad-1; his-3* | *Wiles et al., 2020* | N3756 | *mat A; Sad-1; his-3* |
| Strain, strain background (*N. crassa*) | Δ*set-7* | FGSC# 11182 | N4718 | *mat a; Δset-7::hph* |
| Strain, strain background (*N. crassa*) | Δ*set-7* | *Jamieson et al., 2018* | N4730 | *mat A; Δset-7::bar* |
| Strain, strain background (*N. crassa*) | *ash1*$^{Y888F}$ | *Bicocca et al., 2018* | N4878 | *mat A; his-3; ash1*$^{Y888F}$*::3xFLAG::hph* |
| Strain, strain background (*N. crassa*) | *pNCU07152::nat-1; Δset-7* | *Wiles et al., 2020* | N5807 | *Mat A; pNCU07152::nat-1; Δset-7::bar* |
| Strain, strain background (*N. crassa*) | *pNCU07152::nat-1* | *Wiles et al., 2020* | N5808 | *mat a; pNCU07152::nat-1* |
| Strain, strain background (*N. crassa*) | Δ*iswi* | FGSC 11780 | N6170 | *mat A; Δiswi::hph* |
| Strain, strain background (*N. crassa*) | Δ*iswi* | This study | N6171 | *mat a; Δiswi::hph* |
| Strain, strain background (*N. crassa*) | Mutant hunt strain | *Wiles et al., 2020* | N6279 | *mat a; pNCU05173::hph; pNCU07152::nat-1; his-3* |

*Continued on next page*

*Continued*

| Reagent type (species) or resource | Designation | Source or reference | Identifiers | Additional information |
|---|---|---|---|---|
| Strain, strain background (N. crassa) | $iswi^{L430P}$ original mutant | This study | N6606 | mat a; pNCU05173::hph; pNCU07152::nat-1; his-3; $iswi^{L430P}$ |
| Strain, strain background (N. crassa) | pNCU07152::nat-1; Δiswi::hph | This study | N6727 | mat a; pNCU07152::nat-1; Δiswi::hph |
| Strain, strain background (N. crassa) | $ash1^{Y888F}$ | This study | N6876 | mat a; $ash1^{Y888F}$::3xFLAG::nat-1 |
| Strain, strain background (N. crassa) | $ash1^{Y888F}$ | This study | N6877 | mat a; $ash1^{Y888F}$::3xFLAG::nat-1 |
| Strain, strain background (N. crassa) | EPR-1-Dam | **Wiles et al., 2020** | N7525 | mat A; epr-1::10xGly::Dam::nat-1 |
| Strain, strain background (N. crassa) | EPR-1-Dam; Δeed | **Wiles et al., 2020** | N7538 | mat a; epr-1::10xGly::Dam::nat-1; Δeed::hph |
| Strain, strain background (N. crassa) | Free-Dam; Δset-7 | This study | N7476 | mat A; Δset-7::hph;his-3$^+$::NLS(SV40)::Dam::3xFLAG::nat-1 |
| Strain, strain background (N. crassa) | Free-Dam; Δset-7 | This study | N7477 | mat a; Δset-7::hph;his-3$^+$::NLS(SV40)::Dam::3xFLAG::nat-1 |
| Strain, strain background (N. crassa) | Free-Dam | This study | N7802 | mat A; his-3$^+$::NLS(SV40)::Dam::3xFLAG::nat-1 |
| Strain, strain background (N. crassa) | $iswi^{L430P}$ complement-ation strain | This study | N7810 | mat a; pNCU05173::hph; pNCU07152::nat-1; his-3$^+$::P$_{ccg-1}$::3xFLAG::iswi$^{WT}$; $iswi^{L430P}$ |
| Strain, strain background (N. crassa) | pNCU07152::nat-1; Δiaf-2 | This study | N7941 | mat a; pNCU07152::nat-1; Δiaf-2::hph |
| Strain, strain background (N. crassa) | $acf1^{D161fs}$ original mutant | This study | N7953 | mat a; pNCU05173::hph; pNCU07152::nat-1; his-3; $acf1^{D161fs}$ |
| Strain, strain background (N. crassa) | pNCU07152::nat-1; Δacf1 | This study | N7956 | mat a; pNCU07152::nat-1; Δmus-52::bar Δacf1::hph |
| Strain, strain background (N. crassa) | pNCU07152::nat-1; Δiaf-3::hph | This study | N7960 | mat A; pNCU07152::nat-1; Δiaf-3::hph |
| Strain, strain background (N. crassa) | pNCU07152::nat-1; Δiaf-1:hph | This study | N7961 | mat a; pNCU07152::nat-1; Δiaf-1:hph |
| Strain, strain background (N. crassa) | Δiaf-3 | This study | N7966 | mat A; Δiaf-3::hph |
| Strain, strain background (N. crassa) | ACF1-HA | This study | N7971 | mat a; Δmus-52::bar acf1::HA::hph |

*Continued on next page*

*Continued*

| Reagent type (species) or resource | Designation | Source or reference | Identifiers | Additional information |
|---|---|---|---|---|
| Strain, strain background (*N. crassa*) | IAF-1-HA | This study | N7973 | *mat A; Δmus-52::bar; iaf-1*::HA::*hph* |
| Strain, strain background (*N. crassa*) | Δ*iaf-2* | This study | N7988 | *mat a; Δiaf-2::hph* |
| Strain, strain background (*N. crassa*) | Δ*iaf-2* | This study | N7989 | *mat a; Δiaf-2::hph* |
| Strain, strain background (*N. crassa*) | Δ*iaf-1* | FGSC 12715 | N7990 | *mat a; Δiaf-1::hph* |
| Strain, strain background (*N. crassa*) | Δ*iaf-1* | This study | N7992 | *mat a; Δiaf-1::hph* |
| Strain, strain background (*N. crassa*) | Δ*acf1* | This study | N8016 | *mat a; Δacf1::hph* |
| Strain, strain background (*N. crassa*) | Δ*acf1* | This study | N8017 | *mat a; Δacf1::hph* |
| Strain, strain background (*N. crassa*) | Δ*iaf-3* | This study | N8018 | *mat A; Δiaf-3::hph* |
| Strain, strain background (*N. crassa*) | IAF-3-HA | This study | N8071 | *mat A; pNCU07152::nat-1; iaf-3*::HA::*hph* |
| Strain, strain background (*N. crassa*) | IAF-2-HA | This study | N8075 | *mat a; pNCU07152::nat-1; iaf-2*::HA::*hph* |
| Strain, strain background (*N. crassa*) | ACF1-Dam; Δ*set-7* | This study | N8113 | *mat A; Δset-7::hph; Δmus-52::bar acf1*::Dam::*nat-1* |
| Strain, strain background (*N. crassa*) | ACF1-Dam; Δ*set-7* | This study | N8114 | *mat a; Δset-7::hph; Δmus-52::bar acf1*::Dam::*nat-1* |
| Strain, strain background (*N. crassa*) | ACF1-Dam | This study | N8115 | *mat A; Δmus-52:bar acf1*::Dam::*nat-1* |
| Strain, strain background (*N. crassa*) | *acf1*$^{D161fs}$complement-ation strain | This study | N8142 | *mat a; pNCU05173::hph; pNCU07152::nat-1; his-3$^{+}$::P$_{ccg-1}$::acf1$^{WT}$*::mCherry; *acf1$^{D161fs}$* |
| Strain, strain background (*N. crassa*) | ACF1-Dam | This study | N8146 | *mat a; Δmus-52::bar acf1*::Dam::*nat-1* |
| Strain, strain background (*N. crassa*) | *pNCU07152::nat-1;* Δ*hfp-1* | This study | N8197 | *mat a; pNCU07152::nat-1; Δhfp-1::hph* |
| Sequence-based reagent | hH4_qPCR_FP (4082) | *Jamieson et al., 2013* | ChIP-qPCR primer | CATCAAGGGGTCATTCAC |
| Sequence-based reagent | hH4_qPCR_RP (4083) | *Jamieson et al., 2013* | ChIP-qPCR primer | TTTGGAATCACCCTCCAG |

*Continued on next page*

*Continued*

| Reagent type (species) or resource | Designation | Source or reference | Identifiers | Additional information |
|---|---|---|---|---|
| Sequence-based reagent | NCU07152_promoter_FP (6565) | *Wiles et al., 2020* | ChIP-qPCR primer | CGGTTCCAAAACTGCCCCTGTG |
| Sequence-based reagent | NCU07152_promoter_RP (6645) | *Wiles et al., 2020* | ChIP-qPCR primer | CTCAGCGGGGTATATCAACGGC |
| Sequence-based reagent | NCU05173_promoter_FP (6567) | *Wiles et al., 2020* | ChIP-qPCR primer | GCATTACCCTCGACAGGGTCTG |
| Sequence-based reagent | NCU05173_promoter_RP (6646) | *Wiles et al., 2020* | ChIP-qPCR primer | GCTACCACCATGTGAAGCTCTGG |
| Sequence-based reagent | *his-3*_FP (1665) | *Klocko et al., 2019* | Southern probe primers | GACGGGGTAGCTTGGCCCTAATTAACC |
| Sequence-based reagent | *his-3*_RP (3128) | *Klocko et al., 2019* | Southern probe primers | CGATTTAGGTGACACTATAG |
| Sequence-based reagent | Tel_VIIL_FP (5271) | *Wiles et al., 2020* | Southern probe primers | GGCATCCGTGGGTGTCCCAG |
| Sequence-based reagent | Tel_VIIL_RP (5272) | *Wiles et al., 2020* | Southern probe primers | TTCCCGTCCCTACCAGGCAT |
| Sequence-based reagent | *NCU05173*_FP (6567) | *Wiles et al., 2020* | Southern probe primers | GCATTACCCTCGACAGGGTCTG |
| Sequence-based reagent | *NCU05173*_RP (6568) | *Wiles et al., 2020* | Southern probe primers | CCTGTTCGAGTTATCGGTGTTG |
| Antibody | α-H3K27me2/3 (mouse monoclonal) | Active Motif | Cat. #39536 | Chromatin immunoprecipitation (2 µl ChIP-seq; 3 µl ChIP-qPCR) |
| Antibody | α-H3K36me2 (rabbit polyclonal) | Abcam | Cat. #ab9049 | Chromatin immunoprecipitation (2 µl) |
| Antibody | α-H3K36me3 (rabbit polyclonal) | Abcam | Cat. #ab9050 | Chromatin immunoprecipitation (2 µl) |
| Antibody | α-HA (mouse monoclonal) | MBL | Cat. #180-3 | Immunoprecipitation (20 µl) |
| Antibody | α−FLAG M2 affinity gel (mouse monoclonal) | Sigma-Aldrich | Cat. #A2220 | Immunoprecipitation (400 µl) |
| Peptide, recombinant protein | HA peptide | Thermo Fisher Scientific | Cat. #26184 | Elution |
| Peptide, recombinant protein | 3× Flag peptide | APExBIO | Cat. #A6001 | Elution |

## Strains, media, and growth conditions

All *N. crassa* strains were grown as previously described (*Wiles et al., 2020*) and are listed in the Key resources table. Technical replicates are defined as experimental repeats with the same strain. Biological replicates are defined as experiments performed using a different strain with the same genotype.

## Selection for mutants defective in Polycomb silencing

The selection was carried out as previously described (*Wiles et al., 2020*). Briefly, conidia from strain N6279 were mutagenized with UV radiation and subjected to selection with Hygromycin B or Nourseothricin. Resistant colonies were grown and crossed to strain N3756 to generate homokaryons.

## Whole-genome sequencing, mapping, and identification of mutants

Whole-genome sequencing, SNP mapping, and identification of mutants were performed as previously described (*Wiles et al., 2020*). Briefly, antibiotic-resistant, homokaryotic mutants were crossed

to a genetically polymorphic Mauriceville strain and approximately 15–20 antibiotic-resistant progeny were pooled and prepared for whole-genome sequencing using the Nextera Kit (Illumina, FC-121-1030). Mapping of the critical mutations was performed as previously described (*Hunter, 2007*; *Pomraning et al., 2011*). FreeBayes and VCFtools were used to identify novel genetic variants present in pooled mutant genomic DNA (*Danecek et al., 2011*; *Garrison and Marth, 2012*). All whole-genome sequencing data are available on NCBI Sequence Reads Archive (PRJNA714693).

## Immunoprecipitation followed by MS

Strains N7810 (*his-3::P_{ccg}::3xFLAG-iswi*), N7971 (endogenous *acf1*-HA), N8071 (endogenous *iaf-3*-HA), N7973 (endogenous *iaf-1*-HA), and N8075 (endogenous *iaf-2*-HA) were grown and protein extracted as previously described (*McNaught et al., 2020*) except that 500ml cultures were used. Purification of 3×FLAG-tagged protein was performed as previously described (*McNaught et al., 2020*). For HA-tagged proteins, the same procedure was used except that 20 µg of α-HA antibody (MBL 180-3) was bound to 400-µl equilibrated Protein A agarose (Invitrogen, 15918014) by rotating at room temperature for 1 hr and washed 3× with extraction buffer and protein was eluted 3× with 300µl of 1 mg/ml HA peptide (Thermo Fisher Scientific, 26184) in 1× TBS. Samples were sent to and processed by the UC Davis Proteomics Core Facility for MS and analysis.

## RNA isolation, RT-qPCR, and mRNA-seq

Total RNA was extracted from germinated conidia as previously described (*Wiles et al., 2020*) and used for mRNA-seq library preparation (*Klocko et al., 2016*). Sequencing was performed by the University of Oregon Genomics and Cell Characterization Core Facility.

## mRNA-seq data analysis

Sequence reads were aligned to the *N. crassa* genome (OR74A) using STAR program (version 2.7.3a). Total aligned reads per *N. crassa* gene were calculated using RSEM software (version 1.3.1) and normalized using DESeq2 software (version 1.24.0). Batch effects were corrected using R package, limma (version 3.44.1). FDR <0.05 and abs(log2 fold change)>2 were used as a threshold to identify significantly up- or downregulated genes.Clustered heatmaps were generated using all H3K27-methylated genes that had reads in the mRNA-seq data sets (n=821). Genes were sorted by giving the highest priority to common upregulated genes. If the number of common gene sets was the same, they were prioritized by genes upregulated in the following order: Δ*iswi*, *ash1*^Y888F^, Δ*acf1*, Δ*set-7*, and Δ*epr-1*. Genes were further sorted by $\log_2$ fold change value. All sequencing files are available on the NCBI GEO database (GSE168277).

## ChIP, ChIP-qPCR, and ChIP-seq

H3K27me2/3 ChIP using α-H3K27me2/3 antibody (Active Motif, 39536), which recognizes di- or trimethylated H3K27, was performed as previously described (*Wiles et al., 2020*). H3K36me2 and H3K36me3 ChIP using α-H3K36me2 (Abcam, ab9049) and α-H3K36me3 (Abcam, ab9050) antibodies were performed as previously described (*Bicocca et al., 2018*). The isolated DNA was used for qPCR (see Key resources table for primers) or prepared for sequencing (*Wiles et al., 2020*). Sequencing was performed by the University of Oregon Genomics and Cell Characterization Core Facility.

## ChIP-seq data analysis

Mapping, visualization, and analysis of ChIP-sequencing reads was performed as previously described (*Wiles et al., 2020*). H3K27me2/3 ChIP-seq tracks were normalized using RPKM and H3K36me3 tracks were normalized to 10 million reads using HOMER (*Heinz et al., 2010*). To generate scatter plots for H3K27me2/3, H3K36me2, and H3K36me3 in Δ*iswi* and Δ*acf1* strains ChIP-seq normalized scores were calculated using HOMER to normalize the total tag to 10 M. Bam files from replicates were merged using 'samtools,' then normalized using HOMER to make mixed data. 'bigWigAverageOverBed' from kentUtils was used to generate the average at each gene. Differentially enriched genes (p-value <0.05 and $\log_2$ fold change>1 for gains and <1 for losses) were defined using edgeR package of R (version 3.30.3). Average scores from two replicates were used for the analysis. Average scores from the mix were used for the scatter plot. All sequencing files are available on the NCBI GEO database (GSE168277).

## DamID Southern hybridization and sequencing

Southern hybridization was carried out as previously described (*Miao et al., 2000*) with probes generated by PCR amplification (see Key resources table for primers) from wild-type *N. crassa* genomic DNA (*NCU05173*, TelVIIL) or plasmid pBM61 (*his-3*). Genomic DNA was prepared for DamID-seq as previously described (*Zhou, 2012*) with the modifications we have reported (*Wiles et al., 2020*). Sequencing was performed by the University of Oregon Genomics and Cell Characterization Core Facility. The 'Free Dam' strain had an N-terminal NLS (SV40) and a C-terminal 3× FLAG tag and was expressed from the *his-3* locus.

## DamID-seq data analysis

DamID-seq mapping and analysis were done using the Galaxy public server (*Afgan et al., 2018*). The Barcode Splitter was used to filter for reads with a GATC at the 5′ end and these reads were mapped using Bowtie2 (*Langmead and Salzberg, 2012*). Files for biological replicates were merged using MergeBam. Merged bam files were used as input for bamCoverage (RPKM, 50-bp bins) to generate bigwig files for viewing on IGV and running bigwigCompare. The output from bamCoverage was used with computeMatrix to generate files to use for plotProfile and output graphs. All sequencing files are available on the NCBI GEO database (GSE168277).

## MNase digestion and sequencing

*N. crassa* cells were grown and digested with micrococcal nuclease as previously described (*McKnight et al., 2021*) with the following modifications. MNase (Takara) concentration was optimized for each strain to yield ~80%–90% mononucleosomes (20 units for N3752, N3753, N7966, N8018, N7990, N7992, N7988, and N7989; 40 units for N6877; 60 units for N4718; and 80 units for N4730, N6170, N6171, N6876, N8016, and N8017). All digestions were for 10 min at 37°C, RNase (40 μg) treatment was for 1.5 h at 42°C, and proteinase K (200 μg) treatment was for 1 hr at 65°C. About 10 μg of gel-purified mononucleosome DNA was prepared for high-throughput sequencing using the NEBNext DNA Library Prep Master Mix Set for Illumina (NEB). Sequencing was performed by the University of Oregon Genomics and Cell Characterization Core Facility.

## MNase-seq data analysis

Paired-end sequence reads were aligned to the *N. crassa* genome (OR74A) using Bowtie2 (version 2.3.3) with the option '-q -p 4X 250 `--no-discordant --no-mixed --no-unal`.' Paired-end alignment reads with maximum 250-bp distance gap between them were used in subsequent analysis. This length corresponds to mononucleosomes. Only correctly aligned paired-end alignment reads were filtered using samtools (version 1.5) commands 'samtools view –hf 0x2 input.bam | grep –v "XS:i:"' Dyad Coverage was calculated using the scripts (03_PNA_SDE.R) (*Baldi et al., 2018*). All sequencing files are available on the NCBI GEO database (GSE168277).

## Spectral density estimation

The spectral density (SD) score corresponding to periods of 182 bp was calculated using the scripts (cov2spec.R) (*Baldi et al., 2018*). SD score was normalized as Z-score: (log2(SD score)−average)/standard deviation. Regions with the average Z-score threshold of 2 were defined as the domain with a regular nucleosome array.

## Autocorrelation function

The autocorrelation function (*Braunschweig et al., 2009*) was calculated for the dyad coverage vectors for the lag length of 1000 bp. Nucleosome repeat lengths were obtained by linear regression of the first and second autocorrelation peak positions with zero intercept. The slope of the regression was defined as repeat length.

## Estimation of +1 nucleosome position

The average score of dyad coverage vector for every 182 bp using the region –100 bp to +1000 bp from TSS was calculated for each gene. The closest peak from TSS was defined as +1 nucleosome position.

## Acknowledgements

The authors thank J Lyle and R Morse for help in genetic mapping of UV-generated mutants; V Bicocca for performing initial experiments characterizing ISWI and for comments on the manuscript; and J McKnight and L McKnight for providing guidance and reagents for rapid MNase digestion. The authors also thank T Bailey, D Donovan, L McKnight, and K Noma for helpful comments on the manuscript. This work was funded by the National Institute of General Medical Sciences (GM127142 and GM093061 to EUS), American Heart Association (14POST20450071 to ETW), and KJM was partially supported by the National Institutes of Health (HD007348).

## Additional information

### Competing interests

Kevin J McNaught: is affiliated with Genapsys, Inc. The author has no financial interests to declare. The other authors declare that no competing interests exist.

### Funding

| Funder | Grant reference number | Author |
| --- | --- | --- |
| National Institutes of Health | GM127142 | Eric U Selker |
| National Institutes of Health | GM093061 | Eric U Selker |
| American Heart Association | 14POST20450071 | Elizabeth T Wiles |
| National Institutes of Health | HD007348 | Kevin J McNaught |

The funders had no role in study design, data collection and interpretation, or the decision to submit the work for publication.

### Author contributions

Elizabeth T Wiles, Conceptualization, Data curation, Formal analysis, Investigation, Methodology, Resources, Supervision, Validation, Writing – original draft; Colleen C Mumford, Data curation, Investigation, Validation, Writing – review and editing; Kevin J McNaught, Investigation, Resources, Validation, Writing – review and editing; Hideki Tanizawa, Conceptualization, Data curation, Formal analysis, Methodology, Software, Validation, Writing – review and editing; Eric U Selker, Conceptualization, Formal analysis, Funding acquisition, Investigation, Methodology, Project administration, Resources, Supervision, Writing – review and editing

### Author ORCIDs

Elizabeth T Wiles http://orcid.org/0000-0001-7269-7466
Colleen C Mumford http://orcid.org/0000-0003-0213-0901
Kevin J McNaught http://orcid.org/0000-0002-6887-3161
Hideki Tanizawa http://orcid.org/0000-0002-2573-2473
Eric U Selker http://orcid.org/0000-0001-6465-0094

### Decision letter and Author response

Decision letter https://doi.org/10.7554/eLife.77595.sa1
Author response https://doi.org/10.7554/eLife.77595.sa2

## Additional files

### Supplementary files

• Transparent reporting form

## Data availability

All RNA-seq, ChIP-seq, DamID-seq and MNase-seq data generated in this study have been submitted to the NCBI Gene Expression Omnibus (GEO; https://www.ncbi.nlm.nih.gov/geo/) under accession number GSE168277. All whole genome sequencing data haven been submitted to the NCBI Sequence Read Archive (SRA, https://www.ncbi.nlm.nih.gov/sra) under accession number PRJNA714693.

The following datasets were generated:

| Author(s) | Year | Dataset title | Dataset URL | Database and Identifier |
|-----------|------|---------------|-------------|-------------------------|
| Selker EU | 2022 | The ACF chromatin remodeling complex is essential for Polycomb repression | http://www.ncbi.nlm.nih.gov/geo/query/acc.cgi?acc=GSE168277 | NCBI Gene Expression Omnibus, GSE168277 |
| Selker EU | 2022 | The ACF chromatin remodeling complex is essential for Polycomb repression | https://www.ncbi.nlm.nih.gov/bioproject/PRJNA714693 | NCBI BioProject, PRJNA714693 |

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
