## [Editor Report]

This manuscript provides strong evidence that ACF directly functions to promote Polycomb-dependent repression through chromatin remodeling, which has not been demonstrated. In addition, PRC2/H3K27me-dependent ACF targeting is novel. Finally, the authors' model that facultative chromatin can be classified into several groups based on their dependence on SET7, ASH1, and ACF (Figure 6) is potentially important for guiding future research directions of the field.

---

## [Decision Letter]

**Decision letter after peer review:**

[Editors’ note: the authors submitted for reconsideration following the decision after peer review. What follows is the decision letter after the first round of review.]

Thank you for submitting the paper "The ACF Chromatin Remodeling Complex is Essential for Polycomb Repression" for consideration by *eLife*. Your article has been reviewed by 3 peer reviewers, including Jerry L Workman as the Reviewing Editor and Reviewer #1, and the evaluation has been overseen by a Senior Editor.

The three reviewers discussed the manuscript in depth. The reviewers all felt that the manuscript is interesting and contains novel findings. Other aspects of the manuscript seemed somewhat underdeveloped and needed additional experimentation. In particular, the reviewers wanted to see the effect of the mutants on H3K36 methylation, more evidence that nucleosome repositioning was responsible for derepression and the dependence of nucleosome positioning by ISWI on H3K27 methylation. As it is anticipated that these experiments will require more time than allowed per *eLife* policy for revision, we will not further consider this version of the manuscript, but we would be to consider a new version if the reviewers' comments can be addressed.

*Reviewer #1:*

Polycomb repression of heterochromatic genes differs in different organisms but has been most widely studied in *Drosophila*. Neurospora lacks components of *Drosophila* polycomb repression complexes.

Using a powerful forward genetic screen the authors found that components of the ACF complex were required to maintain repression of H3K27 methylated heterochromatic genes in Neurospora.

ACF binds widely to chromatin across the genome and is not restricted to heterochromatic genes. This indicates that it also functions outside of heterochromatin. Its interaction with heterochromatin is affected somewhat by the loss of H3K27 methylation.

ACF appears to be necessary to position the +1 nucleosome over the promoter of H3K27 methylated heterochromatic genes.

1. "We found that while the majority of gene expression changes observed upon loss of ISWI or ACF1 occurred outside of H3K27-methylated domains" Do they authors suggest that ACF performs a specialized function on H3K27 methylated domains or that it does the same function on all genes with a different outcome on H3K27 methylated genes? Please clarify.

2. A table presenting at least the relative parts of the mass spec data needs to be included in the manuscript so the readers can judge the spectrum counts relevance.

3. Figure 3A is a strange representation of ms data and probably should be removed. The table above should present that data.

4. The authors should limit their conclusions about Iswi protein complexes etc. Without more biochemistry we don't know how many complexes there are, and which components are in each. This would require co-fractionation experiments and probably more purifications. The authors can conclude that the interactions they detect by ms are consistent with iswi being in multiple complexes similar to that described in other organisms.

*Reviewer #2:*

In this paper Wiles et al. show that mutations in the iswi and acf genes, which encode components of a nucleosome remodeling complex, lead to expression of a subset of H3K27me-repressed genes. The strengths of the paper include the detailed genomic analysis supporting the statements that Iswi and Acf regulate a subset of H3K27me3-repressed genes. Data showing that the +1 nucleosome shifts 50bp in H3K27me-genes upregulated in the iswi mutant is also very strong. There is strong data documenting the proteins that Iswi interacts with in *N. crassa*. The data showing the nucleosome shift in the acf mutant is not as strong. The summary figure is highly speculative because there is no data for discrete localization of Acf. Another piece of data that is lacking is what happens to H3K36me in iswi and acf mutants. Knowing this is important because a similar set of genes seem to be derepressed in an ash1 mutant as in the acf and iswi mutants, although the level of depression in ash1 is not as great as in iswi mutants. The summary diagram shows loss of H3K36me as a separate mechanism than loss of the ACF complex. We don't know that since there was no analysis of H3K36me in iswi or acf mutants. Still, the major findings of the paper are important.

1) The paper would be strengthened by an examination of H3K36me in an iswi mutant.

2) Please comment more extensively on why the iswi mutant phenotype is so much stronger than the acf phenotype.

3) How do you think Acf gets localized to H3K27me domains?

4) In Figure 4 you show by qPCR that H3K27me is still present at the two genes used in the initial mutant screen. Please show additional examples of the H3K27me levels on other upregulated genes in the iswi and acf mutants. Are the H3K27me levels altered in any of these genes, or did I miss this?

*Reviewer #3:*

This manuscript clearly shows that, in Neurospora, ACF chromatin remodeler represses some of H3K27-methylated genes in an H3K27me-dependent fashion. This is the first report to demonstrate ISWI class remodeler functions through H3K27me. The main weakness of the manuscript is the modest impact on mechanistic understanding of how ACF or H3K27me functions. Also, the authors' model needs additional support to firmly establish the causality between ACF-dependent nucleosome repositioning and transcriptional repression.

In this manuscript, the authors showed that the ACF chromatin remodeler in *N. crassa* is required for repression of H3K27-methylated genes. The authors further demonstrated that ACF is targeted to these genes through H3K27me, and proposed a model in which ACF-dependent nucleosome repositioning around TSSs causes transcriptional repression. Molecular mechanisms underlying transcriptional repression at facultative heterochromatin is an important topic, and the authors provided convincing data that ACF plays critical roles through H3K27me, which has not been directly demonstrated for any ISWI class remodelers. On the other hand, this manuscript does not provide much advancement on molecular mechanism. Even if the authors' model that ACF-dependent nucleosome repositioning causes transcriptional repression is correct (see below), that mechanism has been described in budding yeast. Therefore, the mechanistic novelty is the H3K27-dependent ACF targeting.

Another issue in the authors' model that ACF-dependent nucleosome repositioning is the cause of transcriptional derepression. As far as I can tell, the data are also equally consistent with the possibility that ACF loss causes transcriptional derepression through other means, and the nucleosomes around TSSs reposition as a result of transcriptional derepression. The authors need to establish the causality more firmly. For example, are there loci where ACF repositions nucleosomes around TSSs similarly to other targets, but transcription is not derepressed upon ACF mutations? If there are many genes that fall into category, that would be consistent with the authors' model. On the other hand, if the authors look at genes that are not direct ACF targets but are derepressed similarly to ACF targets upon ACF mutations, do they see nucleosome repositioning similar to ACF targets? If not, that would argue transcriptional derepression is not sufficient to reposition nucleosomes around TSSs, supporting the authors' model.

1. Line 122: What is an "early condition phenotype"?

2. Lines 233-238: What is the conclusion drawn from this data?

3. Figure 2: It is interesting that transcriptional de-repression is much more robust in iswi mutant than acf1. Given that ISWI purification did not yield other form of ISWI complexes, this is not because of other forms of ISWI-containing remodelers (CHRAC also depends on ACF1). So, another possibility is that, in an iswi mutant, ACF1 functions in a dominant negative fashion to derepress genes further. If this is the case, iswi acf1 double mutants should behave like an acf1 single mutant. The authors should test this for a few targets genes to establish how ACF functions.

4. Figure 2S1: ~1/3 of genes repressed by Set7 are not repressed by ACF. Do these genes have any special features?

5. Do ACF signals by Dam-ID mapping show something specific at H3K27me genes that are derepressed and/or show changes in nucleosome positioning? If the authors can show specific localization at these genes, it would help support the model that the transcriptional effects mediated by ACF are direct.

6. ISWI regulates only a small subset of H3K27me genes (98 of 836). Are there unique properties of these genes that may contribute to our understanding of their mechanism of derepression? For example, are they regulated by a common transcription factor? Are their nucleosome positions in WT different from H3K27me genes not regulated by ISWI?

[Editors’ note: further revisions were suggested prior to acceptance, as described below.]

Thank you for submitting your article "The ACF chromatin remodeling complex is essential for Polycomb repression" for consideration by *eLife*. Your article has been reviewed by 3 peer reviewers, including Jerry L Workman as the Reviewing Editor and Reviewer #1, and the evaluation has been overseen by Kevin Struhl as the Senior Editor. The following individuals involved in review of your submission have agreed to reveal their identity: Jerry L Workman (Reviewer #1).

In this re-submitted manuscript, the authors addressed issues raised by the reviewers. As a result, this manuscript was substantially strengthened. The main conclusions from Figures1 through 3 are consistent with those of Kamei et al. (2021). However, this manuscript provides strong evidence that ACF directly functions to promote Polycomb-dependent repression through chromatin remodeling, which has not been demonstrated. In addition, PRC2/H3K27me-dependent ACF targeting is novel. Finally, the authors' model that facultative chromatin can be classified into several groups based on their dependence on SET7, ASH1, and ACF (Figure 6) is potentially important for guiding future research directions of the field.

The data that the effect of ACF on the position of +1 nucleosomes is distinct between H3K27me positive vs negative loci (Figure 5, Figure 5 Supplemental 2) is quite interesting. Is the degree of targeting of ACF and its effect on transcription similar between H3K27me positive and negative genes? If so, this result suggests currently unknown effects of H3K27me on ACF activity. In this case, Figure 5 Supplemental 2 should be move to the main Figure and the result deserves more extensive discussions as this would increase the impact of the manuscript. On the other hand, if the degree of ACF targeting and/or the level of transcription can explain the difference in the +1 nucleosome position, it needs to be stated in the text.

---

## [Author Response]

[Editors’ note: the authors resubmitted a revised version of the paper for consideration. What follows is the authors’ response to the first round of review.]

Reviewer #1:Polycomb repression of heterochromatic genes differs in different organisms but has been most widely studied in *Drosophila*. Neurospora lacks components of *Drosophila* polycomb repression complexes.Using a powerful forward genetic screen the authors found that components of the ACF complex were required to maintain repression of H3K27 methylated heterochromatic genes in Neurospora.ACF binds widely to chromatin across the genome and is not restricted to heterochromatic genes. This indicates that it also functions outside of heterochromatin. Its interaction with heterochromatin is affected somewhat by the loss of H3K27 methylation.ACF appears to be necessary to position the +1 nucleosome over the promoter of H3K27 methylated heterochromatic genes.1. "We found that while the majority of gene expression changes observed upon loss of ISWI or ACF1 occurred outside of H3K27-methylated domains" Do they authors suggest that ACF performs a specialized function on H3K27 methylated domains or that it does the same function on all genes with a different outcome on H3K27 methylated genes? Please clarify.

To address this question, we looked at the nucleosome positions at genes that are upregulated in ∆iswi and ∆acf1 but are not H3K27me (for SD genes that are included in nucleosome shift analysis: n=335; n=146, respectively). We found that there was no observable shift in the +1 nucleosome at these genes. This suggests that ACF is performing a specialized function (moving the +1 nucleosome) at H3K27me genes and that the upregulation seen in these mutants at non-H3K27me genes is either via a different mechanism or are indirect effects. It is also possible that there are other direct targets of ACF, but we are not able to see the nucleosome shift when we look at all non-H3K27me, upregulated genes. We have added two figure panels as well as text to clarify this point (Figure 5 —figure supplement 2C,D).

2. A table presenting at least the relative parts of the mass spec data needs to be included in the manuscript so the readers can judge the spectrum counts relevance.

We have included a table with spectrum counts for relevant proteins as Figure 2 —figure supplement 1 and all mass spec data can be found as source data files for Figure 2.

3. Figure 3A is a strange representation of ms data and probably should be removed. The table above should present that data.

See above. We have also kept this representation as we feel is includes some information that is lacking in the table.

4. The authors should limit their conclusions about Iswi protein complexes etc. Without more biochemistry we don't know how many complexes there are, and which components are in each. This would require co-fractionation experiments and probably more purifications. The authors can conclude that the interactions they detect by ms are consistent with iswi being in multiple complexes similar to that described in other organisms.

We have limited our conclusions regarding the components in putative ISWI complexes. We deleted the following “were also identified in our ISWI pull-downs, albeit below our threshold, and suggest the presence of an *N. crassa* CHRAC complex.”

Reviewer #2:In this paper Wiles et al. show that mutations in the iswi and acf genes, which encode components of a nucleosome remodeling complex, lead to expression of a subset of H3K27me-repressed genes. The strengths of the paper include the detailed genomic analysis supporting the statements that Iswi and Acf regulate a subset of H3K27me3-repressed genes. Data showing that the +1 nucleosome shifts 50bp in H3K27me-genes upregulated in the iswi mutant is also very strong. There is strong data documenting the proteins that Iswi interacts with in *N. crassa*. The data showing the nucleosome shift in the acf mutant is not as strong. The summary figure is highly speculative because there is no data for discrete localization of Acf. Another piece of data that is lacking is what happens to H3K36me in iswi and acf mutants. Knowing this is important because a similar set of genes seem to be derepressed in an ash1 mutant as in the acf and iswi mutants, although the level of depression in ash1 is not as great as in iswi mutants. The summary diagram shows loss of H3K36me as a separate mechanism than loss of the ACF complex. We don't know that since there was no analysis of H3K36me in iswi or acf mutants. Still, the major findings of the paper are important.

We feel that the data showing the nucleosome shift in ∆acf1 are quite strong (Figure 5I). We agree that the model is somewhat speculative but we feel it is useful and have addressed concerns e.g. by performing H3K36me ChIP-seq (more below). Although we were unable to ChIP ACF, we find the combination of the ACF1-DamID and nucleosome shifts specifically at the K27me upregulated genes in ∆acf1 provides good evidence that ACF is acting at and localizing to these genomic locations. We have added H3K36me ChIP data in ∆acf1 and ∆iswi (more below).

1) The paper would be strengthened by an examination of H3K36me in an iswi mutant.

We have added H3K36me2 and H3K36me3 ChIP-seq data for ∆iswi and ∆acf1 to Figure 3. Briefly, we found no changes in H3K36me2 and some loss of H3K36me3 in these strains.

2) Please comment more extensively on why the iswi mutant phenotype is so much stronger than the acf phenotype.

We are not certain why the upregulation at H3K27me2/3 marked genes is higher in ∆iswi than in ∆acf1. We presume that the iswi mutant upregulates more genes than acf1 because ISWI is part of other protein complexes thus leading to indirect effects. To clarify, we have added the following sentences to the discussion.

“We found that nearly all H3K27 methylated genes that are upregulated in ∆acf1 also showed increased expression in ∆iswi, whereas ∆iswi had several uniquely upregulated genes. This is consistent with a model in which ACF1 is required for targeting ACF to chromatin targets but requires ISWI to catalyze nucleosome movement and allow for increased transcription. ISWI is also part of other protein complexes which may lead to direct or indirect upregulation of distinct genes.”

3) How do you think Acf gets localized to H3K27me domains?

ACF may interact with transcription factors to target H3K27me domains. Targeting of ACF by transcription factors been demonstrated in budding yeast (although not H3K27me targeting as this mark is absent in yeast) and is discussed in the Discussion section.

4) In Figure 4 you show by qPCR that H3K27me is still present at the two genes used in the initial mutant screen. Please show additional examples of the H3K27me levels on other upregulated genes in the iswi and acf mutants. Are the H3K27me levels altered in any of these genes, or did I miss this?

We used the RNA-seq and ChIP-seq data to address this question and added two new figure panels (Figure 3E,F) showing a scatter plot of the correlation between gene expression changes and changes in H3K27me2/3 level at all H3K27me2/3-marked genes. We found a substantial portion of genes that are upregulated in ∆iswi and ∆acf1 do not show significant loss of H3K27me2/3. We also added expression data for ∆iswi with corresponding ChIP-seq tracks in Figure 3K.

Reviewer #3:This manuscript clearly shows that, in Neurospora, ACF chromatin remodeler represses some of H3K27-methylated genes in an H3K27me-dependent fashion. This is the first report to demonstrate ISWI class remodeler functions through H3K27me. The main weakness of the manuscript is the modest impact on mechanistic understanding of how ACF or H3K27me functions. Also, the authors' model needs additional support to firmly establish the causality between ACF-dependent nucleosome repositioning and transcriptional repression.In this manuscript, the authors showed that the ACF chromatin remodeler in *N. crassa* is required for repression of H3K27-methylated genes. The authors further demonstrated that ACF is targeted to these genes through H3K27me, and proposed a model in which ACF-dependent nucleosome repositioning around TSSs causes transcriptional repression. Molecular mechanisms underlying transcriptional repression at facultative heterochromatin is an important topic, and the authors provided convincing data that ACF plays critical roles through H3K27me, which has not been directly demonstrated for any ISWI class remodelers. On the other hand, this manuscript does not provide much advancement on molecular mechanism. Even if the authors' model that ACF-dependent nucleosome repositioning causes transcriptional repression is correct (see below), that mechanism has been described in budding yeast. Therefore, the mechanistic novelty is the H3K27-dependent ACF targeting.Another issue in the authors' model that ACF-dependent nucleosome repositioning is the cause of transcriptional derepression. As far as I can tell, the data are also equally consistent with the possibility that ACF loss causes transcriptional derepression through other means, and the nucleosomes around TSSs reposition as a result of transcriptional derepression. The authors need to establish the causality more firmly. For example, are there loci where ACF repositions nucleosomes around TSSs similarly to other targets, but transcription is not derepressed upon ACF mutations? If there are many genes that fall into category, that would be consistent with the authors' model. On the other hand, if the authors look at genes that are not direct ACF targets but are derepressed similarly to ACF targets upon ACF mutations, do they see nucleosome repositioning similar to ACF targets? If not, that would argue transcriptional derepression is not sufficient to reposition nucleosomes around TSSs, supporting the authors' model.

We feel that our finding that H3K27-methylated genes that are upregulated when the H3K27 methyltransferase set-7 is deleted do not have a nucleosome shift suggest that transcriptional upregulation in regions of facultative heterochromatin is not sufficient to result in the +1 nucleosome shift (we have moved this figure panel from the supplementary figures to the main figures – Figure 5M). We have also added new data (Figure 5 —figure supplement 2C,D) showing that non-H3K27 methylated genes that are upregulated in ∆iswi and ∆acf1 do not have a nucleosome shift, again suggesting that transcriptional upregulation is not sufficient to induce a nucleosome shift. These data are consistent with the reviewer’s second scenario (above) where non-H3K27me upregulated genes (and presumably indirect targets) show increased transcription without the +1 nucleosome shift.

1. Line 122: What is an "early condition phenotype"?

We’ve modified the sentence to clarify, as follows.

“We noticed that disruption of these two genes resulted in an early conidiation (production of asexual spores) phenotype that appeared as more dense growth in the spot tests (Figure 1C,D) but this was not accompanied by an increased linear growth rate.”

2. Lines 233-238: What is the conclusion drawn from this data?

We removed the original Figure 4F,G showing the position of H3K27me2/3 gains and losses in ∆iswi and ∆acf1 relative to the telomere as we felt these observations were tangential. The new Figure 3 now shows more relevant information regarding changes in H3K27me2/3 and changes in gene expression.

3. Figure 2: It is interesting that transcriptional de-repression is much more robust in iswi mutant than acf1. Given that ISWI purification did not yield other form of ISWI complexes, this is not because of other forms of ISWI-containing remodelers (CHRAC also depends on ACF1). So, another possibility is that, in an iswi mutant, ACF1 functions in a dominant negative fashion to derepress genes further. If this is the case, iswi acf1 double mutants should behave like an acf1 single mutant. The authors should test this for a few targets genes to establish how ACF functions.

While we are unable to make firm conclusions about ISWI-containing complexes with our mass spec data, we do feel they are consistent with ISWI forming multiple protein complexes, as it does in other organisms (see reviewer 1, comment 4). Therefore the most parsimonious explanation for the more robust transcriptional derepression in ∆iswi is because it is a member of multiple protein complexes. However, we do agree that it is interesting that genes that are upregulated in both iswi and acf1 are much more highly expressed when iswi is deleted. We tested the reviewer’s hypothesis by doing RT-qPCR on three genes that were more robustly upregulated in ∆iswi than ∆acf1 based on mRNAseq (NCU07152, NCU08796, NCU04860) in WT, ∆iswi, ∆acf1 and ∆iswi;∆acf1 double mutant strains. We found that gene expression in the double mutant was robust, similar to the ∆iswi single mutant strain, thus ruling out the dominant negative hypothesis. Because this yielded a negative result we did not include it in the manuscript.

4. Figure 2S1: ~1/3 of genes repressed by Set7 are not repressed by ACF. Do these genes have any special features?

We performed a transcription factor binding site motif analysis (+1 nucleosome +/- 50 bp) using HOMER in the subset of genes that are uniquely upregulated in ∆set-7 (n=23), but the program did not return any motifs predicted to be true positive binding sites.

5. Do ACF signals by Dam-ID mapping show something specific at H3K27me genes that are derepressed and/or show changes in nucleosome positioning? If the authors can show specific localization at these genes, it would help support the model that the transcriptional effects mediated by ACF are direct.

While we are able to detect a significant decrease in the ACF1 DamID signal in ∆set-7 at H3K27 methylated regions compared to non-H3K27 methylated regions of the genome, we are unable to detect a further significant decrease at the subset of H3K27 methylated genes that increase expression and have a nucleosome shift in ∆acf1.

6. ISWI regulates only a small subset of H3K27me genes (98 of 836). Are there unique properties of these genes that may contribute to our understanding of their mechanism of derepression? For example, are they regulated by a common transcription factor? Are their nucleosome positions in WT different from H3K27me genes not regulated by ISWI?

As illustrated in Figure 2 H, iswi regulates 180 (98+82) of H3K27-methylated genes. This is even more than the number that are derepressed by loss of the methyl mark itself (the set-7 mutant causes de-repression of 63 genes) and it is important to consider that some H3K27me-marked predicted genes may be “pseudogenes” or else regulated by unknown processes. We attempted to identify a transcription factor that worked with ACF by looking for a binding motif surrounding the +1 nucleosome (+1 nucleosome +/- 50 bp) in the subset of genes that were upregulated and showed a nucleosome shift in ∆iswi (n=93) and ∆acf1 (n=61). No motif was found to be enriched in these regions. By comparing the WT nucleosome positions in genes that are H3K27 methylated and upregulated in ∆acf1 or ∆iswi (Figure 6I, K) to those that are H3K27 methylated and not upregulated in ∆acf1 and ∆iswi (Figure 6J, L) we do notice some differences. We find that overall nucleosomes tend to be more well positioned (smoother, sharper peaks and valleys) in the genes that are not upregulated in these mutants. We also see a less pronounced NDR in the genes that are not upregulated. While these are interesting observations, it is hard to draw any biological conclusions from these differences.

[Editors’ note: what follows is the authors’ response to the second round of review.]

The reviewers have discussed their reviews with one another, and the Reviewing Editor has drafted this to help you prepare a revised submission.In this re-submitted manuscript, the authors addressed issues raised by the reviewers. As a result, this manuscript was substantially strengthened. The main conclusions from Figures1 through 3 are consistent with those of Kamei et al. (2021). However, this manuscript provides strong evidence that ACF directly functions to promote Polycomb-dependent repression through chromatin remodeling, which has not been demonstrated. In addition, PRC2/H3K27me-dependent ACF targeting is novel. Finally, the authors' model that facultative chromatin can be classified into several groups based on their dependence on SET7, ASH1, and ACF (Figure 6) is potentially important for guiding future research directions of the field.The data that the effect of ACF on the position of +1 nucleosomes is distinct between H3K27me positive vs negative loci (Figure 5, Figure 5 Supplemental 2) is quite interesting. Is the degree of targeting of ACF and its effect on transcription similar between H3K27me positive and negative genes? If so, this result suggests currently unknown effects of H3K27me on ACF activity. In this case, Figure 5 Supplemental 2 should be move to the main Figure and the result deserves more extensive discussions as this would increase the impact of the manuscript. On the other hand, if the degree of ACF targeting and/or the level of transcription can explain the difference in the +1 nucleosome position, it needs to be stated in the text.

We performed further analyses to address the question of whether ACF targeting or level of transcriptional upregulation could explain the difference in ACF activity at genes that are upregulated in *∆acf1* strains with and without H3K27 methylation. We examined the ACF1-DamID signal at the +1 nucleosome (+/- 500bp). Upstream of the +1 nucleosome we found that the level of targeting was similar at genes that were methylated and unmethylated with a slight shift in the peak methylation, while downstream we saw ACF1-Dam targeting was slightly higher in H3K27-methylated regions. These findings were consistent with what we report for all genes (H3K27 methylated *vs.* non-methylated) in Figure 4. The level of transcription (log2FC *∆acf1*/WT) was slightly higher (p=0.041) at H3K27-methylated genes compared to non-methylated genes. We did not find these data convincing enough to warrant inclusion in the manuscript. However, since the difference in ACF activity cannot clearly be ascribed to differences in targeting or transcriptional activity, following your suggestion, we decided to better highlight the data suggesting some specificity for H3K27-methylated regions. Thus, we moved the nucleosome plots showing no shift in non- H3K27 methylated that are upregulated in *∆iswi* and *∆acf1* to the main Figure 5. To make room for this, we relegated two of the three analyses of the accessory factors (*iaf-1* and *iaf-2*) to the supplementary figure (Figure 5 Sup2 panels C&D). We also made a minor change to the conclusion drawn from these data in the Results section and added a sentence regarding this in the Discussion.